# Heterochronic faecal transplantation boosts gut germinal centres in aged mice

Marisa Stebegg [1], Alyssa Silva-Cayetano [1], Silvia Innocentin[1], Timothy P. Jenkins[2], Cinzia Cantacessi [2], Colin Gilbert [3] & Michelle A. Linterman [1]

Ageing is a complex multifactorial process associated with a plethora of disorders, which contribute significantly to morbidity worldwide. One of the organs significantly affected by age is the gut. Age-dependent changes of the gut-associated microbiome have been linked to increased frailty and systemic inflammation. This change in microbial composition with age occurs in parallel with a decline in function of the gut immune system; however, it is not clear whether there is a causal link between the two. Here we report that the defective germinal centre reaction in Peyer's patches of aged mice can be rescued by faecal transfers from younger adults into aged mice and by immunisations with cholera toxin, without affecting germinal centre reactions in peripheral lymph nodes. This demonstrates that the poor germinal centre reaction in aged animals is not irreversible, and that it is possible to improve this response in older individuals by providing appropriate stimuli.

[1] Laboratory of Lymphocyte Signalling and Development, Babraham Institute, Babraham Research Campus, Cambridge CB22 3AT, UK. [2] Department of Veterinary Medicine, Madingley Road, Cambridge CB3 0ES, UK. [3] Biological Services Unit, Babraham Institute, Babraham Research Campus, Cambridge CB22 3AT, UK. Correspondence and requests for materials should be addressed to M.A.L. (email: michelle.linterman@babraham.ac.uk)

One of the major achievements of human endeavour is the extension of lifespan through changes in medical care, diet and sanitation. The consequent upward demographic shift in human age creates a challenge for medical science: how to enable people to age in good health. One of the organs that is significantly affected by age is the gastrointestinal tract and the gut-associated microbiome. The gut microbiota comprises hundreds of different commensal bacterial species, as well as fungi, protozoa and viruses. These commensal microorganisms are essential for health, affecting the functions of multiple bodily systems, such as host metabolism, brain functions and the immune response[1]. Older individuals have age-related alterations in gut microbial composition[2–5], which have been associated with increased frailty[2,4], reduced cognitive performance[6], immune inflammaging[7] and an increased susceptibility to intestinal disorders[8].

What drives these age-associated changes in the gut microbiota remains unknown. The microbiome is shaped by many factors including host genetics, early life events, diet, and the gut immune system[9–11]. While some of these factors remain relatively constant throughout life, the function of the immune system is known to deteriorate with age[12]. This prompts the hypothesis that dysbiosis of the intestinal microbiome in older individuals may be driven by altered cross-talk between the host immune system and the microbiota. The gut immune system can regulate the composition of the microbiome by the production of immunoglobulin A (IgA) antibodies that coat commensal bacteria[13]. In the gastrointestinal tract, IgA antibodies are either produced by short-lived plasma cells in the lamina propria or from plasma cells that arise from germinal centre (GC) reactions in Peyer's patches (PPs)[14,15]. In the lamina propria, plasma cells can be generated with or without T cell help, and typically secrete IgA antibodies that are encoded by germline immunoglobulin genes. In GCs, B cells proliferate and undergo somatic mutation of their immunoglobulin genes. GC B cells which are able to bind antigen with improved affinity after somatic mutation receive positive selection signals from T follicular helper (Tfh) cells and follicular dendritic cells that facilitate their differentiation into long-lived antibody secreting plasma cells that secrete high-affinity IgA[14–16]. Negative regulation of the GC reaction is mediated by suppressive T follicular regulatory (Tfr) cells that limit the output of the GC[17]. Loss of Tfh or Tfr cells[18,19] or the absence of somatic hypermutation in GC B cells[20] results in changes in the gut IgA repertoire which alter the composition of the gut microbiome[18–20]. This suggests that GC-derived IgA antibodies can regulate the commensal microbiome.

The studies described above have established the existence of a relationship between GC reactions and the microbiome. Some of these studies[18–20] indicate clearly that the microbiome is causally influenced by the GC reaction. In the case of the gut-associated defects seen with advancing age in the GC reaction and gut microbiota, however, the direction of causation is unclear. Here, we report that the defective GC reaction in aged mice could be boosted by co-housing with younger animals, by direct faecal transplantation from adult donors and by oral administration of cholera toxin. This demonstrates that the age-dependent defect in the gut GC reaction is not irreversible, but can be corrected by changing the microbiota or by delivery of a bacterial derived toxin.

## Results

**The GC reaction is diminished in aged PPs.** To assess the impact of ageing on PPs we compared the GC reaction at this site in 22-month-old "aged" mice with 3-month-old "adult" C57BL/6 and BALB/c mice. Aged mice of both strains showed a reduction in frequency and number of Bcl6+Ki67+B220+GC B cells (Fig. 1a–e, gating strategy in Supplementary Fig. 1). This was specific to the PPs, as there was no reduction in the proportion of GC B cells in the gut-draining mesenteric lymph nodes (LNs) in the same mice (Supplementary Fig. 2A–E). GC size depends on interactions with T cells: GC B cells receive positive signals from Tfh cells, while Tfr cells negatively regulate their response. The frequency of Tfh cells was not affected by age in the PPs (Fig. 1f–j, gating strategy in Supplementary Fig. 1) in either strain, demonstrating that a reduction of Tfh cell number is not the cause of the diminished GC reaction in the PPs of aged mice. In mesenteric LNs, there was an increase in the number and frequency of Tfh cells in aged mice (Supplementary Fig. 2F–J). In the PPs of 22-month-old mice, Tfr cell frequency was not changed in BALB/c mice (Fig. 1k–m) and was slightly decreased in C57BL/6 mice (Fig. 1n, o), indicating that an increase in the proportion of Tfr cells does not contribute to the decreased magnitude of the GC reaction in the PPs of aged mice. There was an increase in Tfr cell numbers in mesenteric LNs from aged mice in both strains, consistent with previous observations[21] (Supplementary Fig. 2K–O). These data show that there is a decrease in the magnitude of the GC reaction in the PPs of aged mice but that this is not obviously linked with age-associated changes in follicular T cell subset composition.

**The composition of the gut microbiome changes with age.** It is known that the output of the GC reaction can influence the composition of the microbiome[22]. Therefore we sought to understand whether the age-dependent change in the GC reaction of PPs was linked with changes in the microbiota. For this, we assessed the bacterial gut microbiome in C57BL/6 and BALB/c mice by 16S rRNA sequencing of DNA extracted from faecal pellets from 22-month-old "aged" and 3-month-old "adult" mice. Principal Coordinates Analysis (PCoA) showed that age impacts the composition of the microbiome and demonstrated that inter-individual variation of the microbiome increases in aged animals (Fig. 2a, b; Supplementary Fig. 2A). In BALB/c mice six bacterial families were reduced in aged mice, whilst two bacterial families increased in abundance with age (Fig. 1c). This corresponded to reduced gut microbial diversity in aged BALB/c compared with adult mice (Fig. 2d). Interestingly, *Bacteroides acidifaciens* and *Lactobacillus gasseri*, both of which have been shown to induce intestinal IgA production, were not detected in aged BALB/c mice (Supplementary Table 1)[23,24]. Aged C57BL/6 males had increased bacterial diversity (Fig. 2e), which was associated with the increased abundance of 7 bacterial families in these mice (Fig. 2f). Bacterial diversity was not changed in aged C57BL/6 females while two bacterial families were significantly decreased (Supplementary Fig. 2B, C). The abundance of *Bacteroides acidifaciens* and *Lactobacillus gasseri* was not affected in either female or male C57BL/6 aged mice (Supplementary Table 1), suggesting that the age-associated immunological phenotype is not caused by a reduction of these particular species. Both BALB/c and C57BL/6 aged males had an expansion of *Firmicutes* at the expense of *Bacteroidetes* at the phylum level (Fig. 2g, h; Supplementary Fig. 3). This analysis shows that the composition of the gut microbiome changes with age in mice and that these age-dependent changes are also shaped by the sex and genetics of the host.

To determine whether there is a link between the poor GC reaction in the PPs of aged mice and antibody regulation of the microbiome, we examined IgA coating of gut bacteria in aged mice. In both the ileum and colon the frequency of IgA coated bacteria was comparable between 3-month-old adult and 22-month-old aged mice from both the C57BL/6 and BALB/c strains (Supplementary Fig. 4). This could be explained by previous

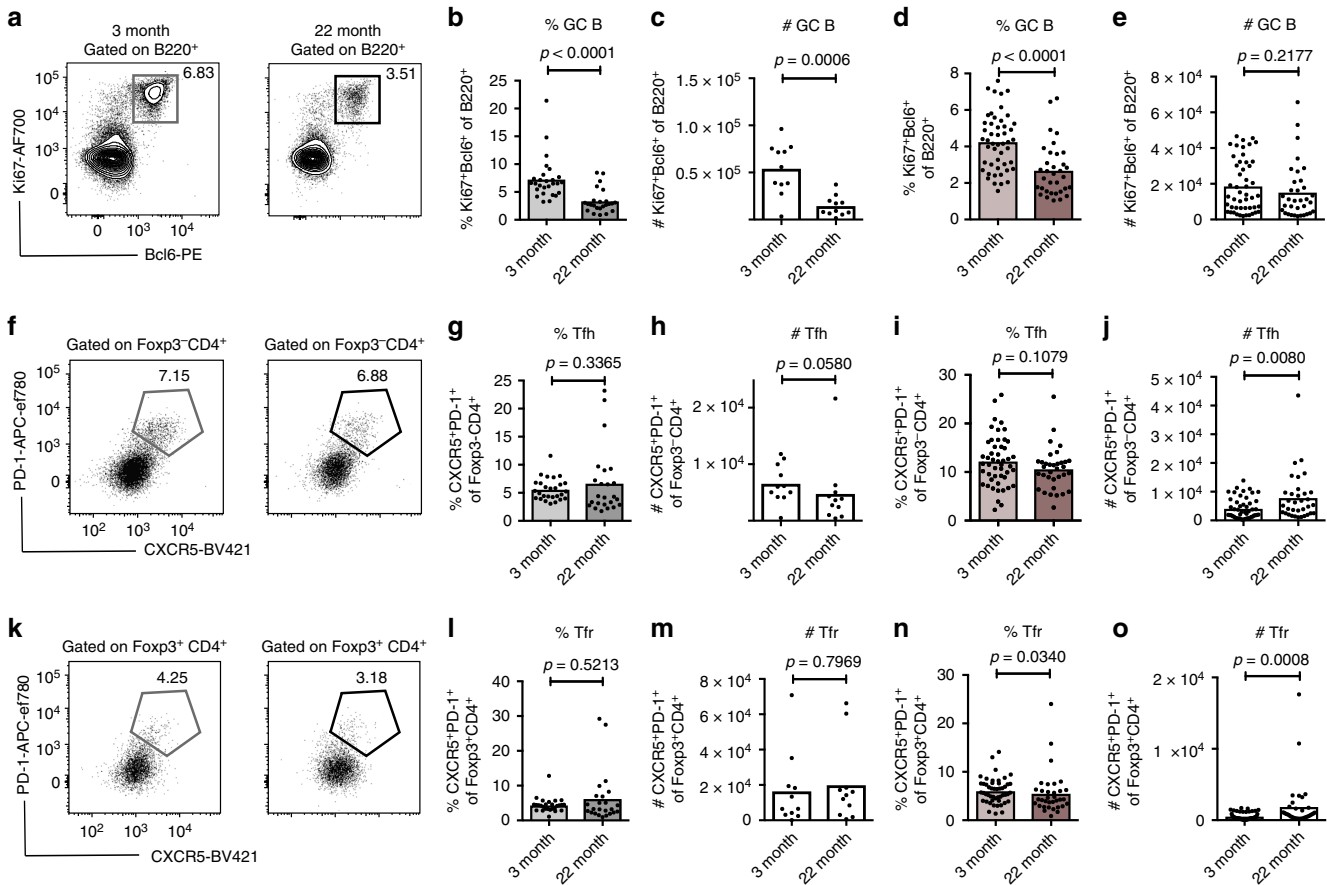

**Fig. 1** Germinal centre B cells are reduced in the Peyer's patches of aged BALB/c and C57BL/6 mice. Flow cytometric analysis of germinal centre (GC) cell populations in the Peyer's patches (PPs) of adult (3-month-old) and aged (22-month-old) BALB/c and C57BL/6 mice. **a–c** Representative flow cytometric plots (**a**) and quantitation of B220+Ki67+Bcl6+ GC B cells (**b**, **c**) in the PPs of 3-month-old and 22-month-old BALB/c mice. **d**, **e** Quantitation of B220+Ki67+Bcl6+ GC B cell percentage (**d**) and number (**e**) in C57BL/6 mice. **f–h** Representative flow plots (**f**) and quantitation of CD4+Foxp3−CXCR5+PD-1+ Tfh cells (**g**, **h**) in the PPs of BALB/c mice. **i**, **j** Quantitation of Tfh cell percentages (**i**) and numbers (**j**) in C57BL/6 mice. **k–m** Representative flow plots (**k**) and quantitation of CD4+Foxp3+CXCR5+PD-1+ Tfr cells (**l**, **m**) in BALB/c mice. **n**, **o** Quantitation of Tfr cell percentages (**n**) and numbers (**o**) in C57BL/6 mice. Bar graphs show the combined results of 3–6 independent experiments which were performed with female BALB/c mice and both male and female C57BL/6 mice with a total of n = 11–49 mice per group. Bar height corresponds to the mean, and each circle represents one biological replicate. P-values were determined using the Mann-Whitney test in GraphPad Prism6. Source data are provided as a Source Data file

studies which described normal levels of intestinal IgA and commensal IgA coating in GC-deficient mice, suggesting that commensal-specific IgA can be produced in a GC-independent manner[25–27]. Taken together, this indicates that the impaired GC reaction in the PPs of aged mice does not influence commensal-reactive IgA, and rather prompts the hypothesis that the composition of the microbiome may influence the magnitude of the GC response.

**Co-housing rescues the reduced PP GC reaction in aged mice.** In our initial experiments, adult and aged mice were housed separately from each other, sharing cages with their respective littermates. Knowing that the gut microbiome can vary between cages[28] and that co-housed mice exchange faecal bacteria by coprophagy[29], we decided to repeat our experiments, using 3-month-old adult and 21-month-old aged mice that were housed in the same cage for 30–40 days. To our surprise, the age-associated reduction of GC B cells in PPs of BALB/c mice was lost upon co-housing (Fig. 3a, b). This correction of the GC reaction was accompanied by an increase in Tfh cells but Tfr cell numbers and IgA coating of commensals were unchanged (Fig. 3c–f; Supplementary Fig. 5). This demonstrates that the age-dependent

deficit in the magnitude of PP GCs is reversible. To determine whether changes in the GC reaction upon co-housing were driven by the transfer of faecal microbiota between mice, bacterial 16S rRNA sequencing was performed on faecal pellets collected before and after co-housing. PCoA revealed that the microbiome of aged mice was more similar to that of 3-month-old adult mice after co-housing, while the microbiome of adult BALB/c mice was not significantly changed (Fig. 3g). Co-housing also increased bacterial Shannon diversity in aged mice to levels similar to those in adult mice (Fig. 3h). This was associated with the detection of bacterial species and families which were not identified in samples from aged mice prior to co-housing (Fig. 3i, j), including the species *Bacteroides acidifaciens* and *Lactobacillus gasseri* (Supplementary Table 2). This rescue of the diminished PP GC reaction in BALB/c mice was replicated in 22-month-old C57BL/6 mice upon co-housing with 3-month-old adult mice (Fig. 4a–f). In C57BL/6 mice, co-housing led to reciprocal microbiota transfer between adult and aged mice, perhaps because there is no age-associated reduction in bacterial diversity in C57BL/6 mice (Fig. 4g, h). Co-housing was associated with a trend for increased bacterial diversity in mice of both ages, although this was not significantly different (Fig. 4h). Taken together, these data suggest that the poor PP GC reaction in aged mice can be rescued by the

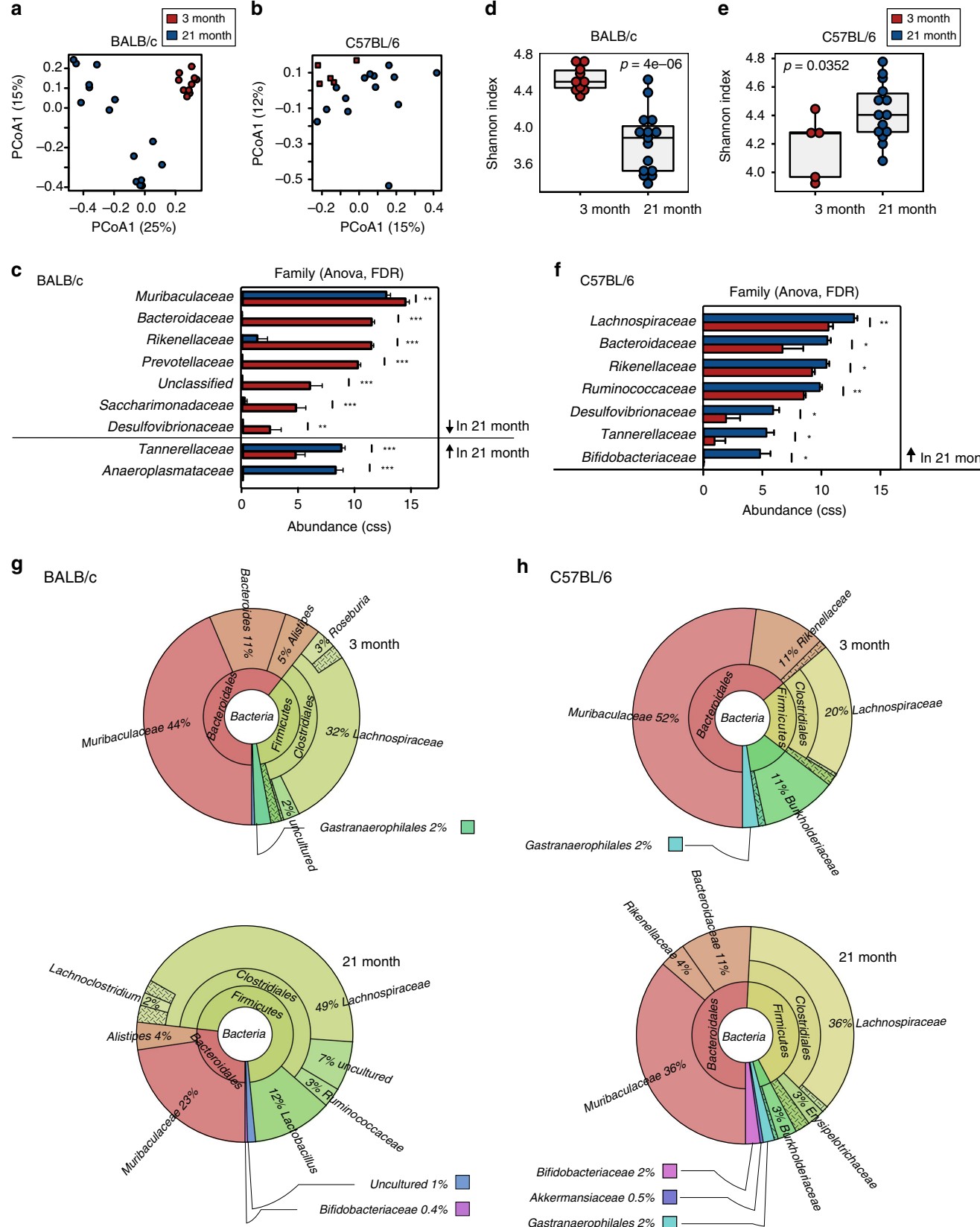

acquisition of the microbiota from younger animals. The rescue of the GC reaction in aged mice occurred independently of genetic background, and there was no overlap between the bacterial families significantly changed by age or co-housing between BALB/c and C57BL/6 mice (Figs. 3j, 4i, Supplementary Table 2).

This suggests that the co-housing-dependent increase of PP GC B cells in aged mice is not driven by a specific bacterial family, but is a response to a comprehensive change in the gut microbiome.

To determine whether co-housing impacts systemic immune responses we assessed the response to subcutaneous NP-KLH

**Fig. 2** The gut microbiome changes during ageing. 16S rRNA sequencing data were generated from faecal pellets collected from adult (3-month-old) and aged (21-month-old) BALB/c females and C57BL/6 males. **a**, **b** Bray-Curtis PCoA and **d**, **e** bacterial diversities (measured by Shannon index) of samples collected from 3-month-old and 21-month-old BALB/c mice (**a**, **d**) and C57BL/6 mice (**b**, **e**). The overall *p*-value was based on ANOVA tests. **c**, **f** Depiction of bacterial families whose abundance was significantly different between 3-month-old and 21-month-old BALB/c (**c**) and C57BL/6 (**f**) mice as determined by ANOVA analysis after cumulative-sum scaling (CSS). **g**, **h** Krona plots depicting the phylogenetic composition of the gut microbiome in 3-month-old (top) and 21-month-old (bottom) BALB/c (**g**) and C57BL/6 (**h**) mice. The percentages shown are averages of the samples in each age group. Samples were collected in 2 independent experiments for BALB/c mice and 3 independent experiments for C57BL/6 mice with a total of *n* = 5–14 mice per group. *P*-values are based on ANOVA tests *FDR ≤ 0.05, **FDR ≤ 0.01, ***FDR ≤ 0.001. Source data are provided as a Source Data file

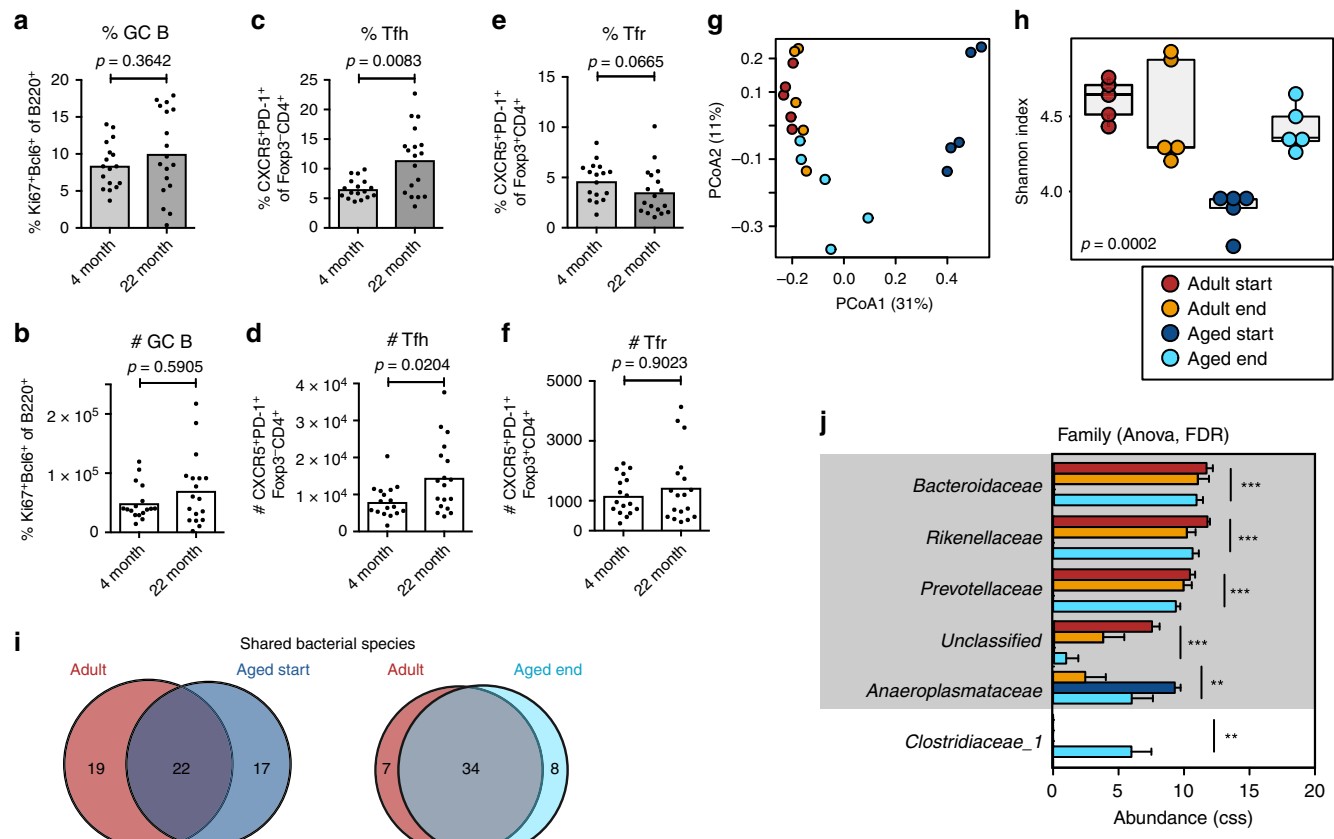

**Fig. 3** Co-housing boosts the gut germinal centre response of aged BALB/c mice. Adult and aged female BALB/c mice were co-housed for 30–40 days, then Peyer's patch (PP) germinal centre (GC) cell populations were analysed by flow cytometry. The percentage and number of B220+Ki67+Bcl6+ GC B cells (**a**, **b**), CD4+Foxp3−CXCR5+PD-1+ Tfh cells (**c**, **d**) and CD4+Foxp3+CXCR5+PD-1+ Tfr cells (**e**, **f**) in Peyer's patches. **g**–**j** 16S rRNA sequencing data were generated from faecal pellets collected from 5 adult and 5 aged BALB/c mice at the start and end of co-housing (*n* = 5 mice per group). Samples were clustered by Bray-Curtis PCoA (**g**) and bacterial diversities were measured by the Shannon index (**h**). The overall *p*-value was based on ANOVA tests. **i** Venn diagrams showing the numbers of shared and unique bacterial species detected in both age groups of mice before co-housing and after co-housing. **j** Depiction of bacterial families whose abundance was significantly different between co-housed BALB/c mice as determined by ANOVA analysis after cumulative-sum scaling (CSS). *FDR ≤ 0.05, **FDR ≤ 0.01, ***FDR ≤ 0.001. Families previously found to be significantly decreased in aged versus adult BALB/c mice before co-housing are marked in grey. In **a**–**f** bar plots show the combined results of 2 independent experiments with a total of *n* = 17–18 mice per group. Bar height corresponds to the mean, and each circle represents one biological replicate. *P*-values were determined using the Mann–Whitney test in GraphPad Prism6. Source data are provided as a Source Data file

immunisation in the draining inguinal LNs in 22-month-old BALB/c mice that were either co-housed with 3-month-old adult mice or with their age-matched littermates. 22-month-old BALB/c mice had a reduced frequency and number of GC B cells in their draining LNs 14 days after immunisation compared with adult mice, irrespective of whether they were cohoused with littermates or with younger animals (Supplementary Fig. 6A–C). Total Tfh cell numbers in the inguinal LNs were not influenced by age or cohousing after immunisation (Supplementary Fig. 6D–E). The reduced GC response in aged mice corresponded with decreased antigen-specific antibody production after vaccination and a

defect in affinity maturation. This diminished antibody response was not influenced by cohousing (Supplementary Fig. 6G–I). Together this indicates that systemic immune responses are not influenced by the microbial transfer that occurs during co-housing, and suggests that the impact of microbial transfer on the GC reaction is limited to the gut.

**Heterochronic faecal transplantation boosts the GC in PPs.** To investigate whether the induction of the GC reaction during co-housing was solely dependent on direct transmission of gut

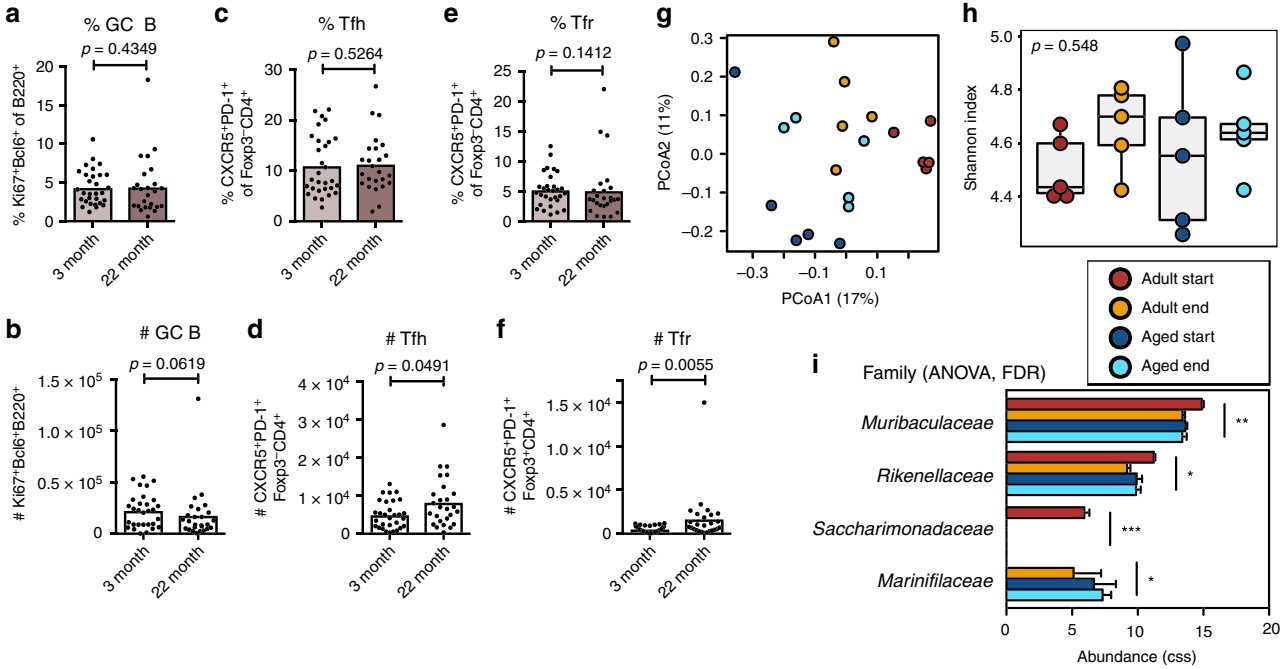

**Fig. 4** Co-housing boosts the gut germinal centre response of aged C57BL/6 mice. **a–f** Adult and aged female C57BL/6 mice were co-housed for 30–40 days, then Peyer's patch (PP) germinal centre (GC) cell populations were analysed. The percentage and number of B220+Ki67+Bcl6+ GC B cells (**a**, **b**), CD4+Foxp3-CXCR5+PD-1+ Tfh cells (**c**, **d**) and CD4+Foxp3+CXCR5+PD-1+ Tfr cells (**e**, **f**) in Peyer's patches as quantified by flow cytometry. 16S rRNA sequencing data were generated from faecal pellets collected from 5 adult 3-month-old and 5 aged C57BL/6 mice before (start) and after (end) co-housing. Bray-Curtis PCoA (**g**) and Shannon diversities (**h**) of samples collected from adult and aged C57BL/6 mice. The overall *p*-value was based on ANOVA testing. **i** Depiction of bacterial families whose abundance was significantly different between co-housed C57BL/6 mice as determined by ANOVA analysis after cumulative-sum scaling (CSS). *FDR ≤ 0.05, **FDR ≤ 0.01, ***FDR ≤ 0.001. In **a–f** bar plots show the combined results of 2 independent experiments with a total of $n = 25$–30 mice per group. Bar height corresponds to the mean, each circle represents one biological replicate. *P*-values were determined using the Mann–Whitney test in GraphPad Prism6. Source data are provided as a Source Data file

microbiota, we conducted faecal microbiota transplantation (FMT) experiments in which recipient mice were given a suspension of faecal pellets from donor mice by oral gavage. The cages were also supplemented with fresh faecal pellets and dirty bedding from donor mice for three weeks (Fig. 5a–h). First, we gavaged 22-month-old C57BL/6 mice with faecal pellets from 3-month-old adult mice (Fig. 5a). Twenty-three days after this treatment, we observed increased GC B cell numbers in aged mice (Fig. 5b, c). We also observed a trend towards increased Tfh cells in aged mice (Fig. 5d, e) and a significant increase in their Tfr cell numbers (Fig. 5f, g). To determine whether the boost in the GC reaction by FMT is exclusive to aged mice, 3-month-old C57BL/6 mice were gavaged with faecal pellets from 22-month-old adult mice (Fig. 5h). For this experiment we used C57BL/6 mice because we had evidence of reciprocal microbial transfer between adult and aged mice of this strain during co-housing: this result indicated that the gut microbiome of both adult and aged C57BL/6 mice is receptive to microbial transfer and presented us with a tool to assess whether the transfer of a new microbiome generally enhances the GC reaction irrespective of age directionality, or if it constitutes a unique feature in aged mice. FMT of younger adult mice with faecal pellets from aged mice led to an increase in GC B, as well as Tfh cells, while Tfr cell numbers were not affected (Fig. 5i–n). This demonstrates that the GC reaction in PPs is highly sensitive to changes in the gut microbiome in both young and aged mice. Furthermore, the boost of the PP GC reaction appears to be independent of the transfer of a specific bacterial family or species, since FMT enhanced the response in both adult and aged recipients.

To determine whether FMT can rescue the PP GC reaction in aged mice of another strain, we performed FMT experiments in 22-month-old BALB/c mice (experimental set-up as in Fig. 5a). GC B and Tfh cells had significantly expanded in 22-month-old mice 23 days after FMT compared with PBS-treated control mice (Fig. 6a–d), but Tfr cells were not changed by FMT treatment (Fig. 6e, f). FMT did not affect the levels of PP-resident IgA+ B cells, free intestinal IgA or bacterial IgA coating in either mouse strain (Supplementary Fig. 7). Bacterial 16S rRNA sequencing confirmed the successful establishment of an "adult" microbiome in 22-month-old mice by FMT (Fig. 6g), with a slight increase in gut microbial diversity in aged mice receiving FMT (Fig. 6h). This included the presence of bacterial species that were originally not detected in aged mice (Fig. 6i; Supplementary Table 3). Thus, microbial transfer is sufficient to restore the defective GC reaction in aged mice. Our data demonstrate that, even though the GCs in PPs diminish during ageing, this age-associated phenotype is not cell-intrinsic and can be rescued by stimulation from the gut microbiome.

**Cross-strain faecal transplantation does not enhance the GC.** The boost of the GC reaction by heterochronic faecal transplantation in both adult and aged mice suggests that alterations in the microbiome can enhance the GC reaction in PPs. To determine whether this occurs independently of an age mismatch between the donor and the recipient, we performed FMT from 3-month-old BALB/c mice into 3-month-old C57BL/6 mice (experimental set-up as in Fig. 5a). This approach was chosen as there are differences in the microbiome between these two strains (Fig. 7a), and adult BALB/c mice have a higher bacterial diversity than C57BL/6 mice (Fig. 7b). Twenty-three days after FMT, there

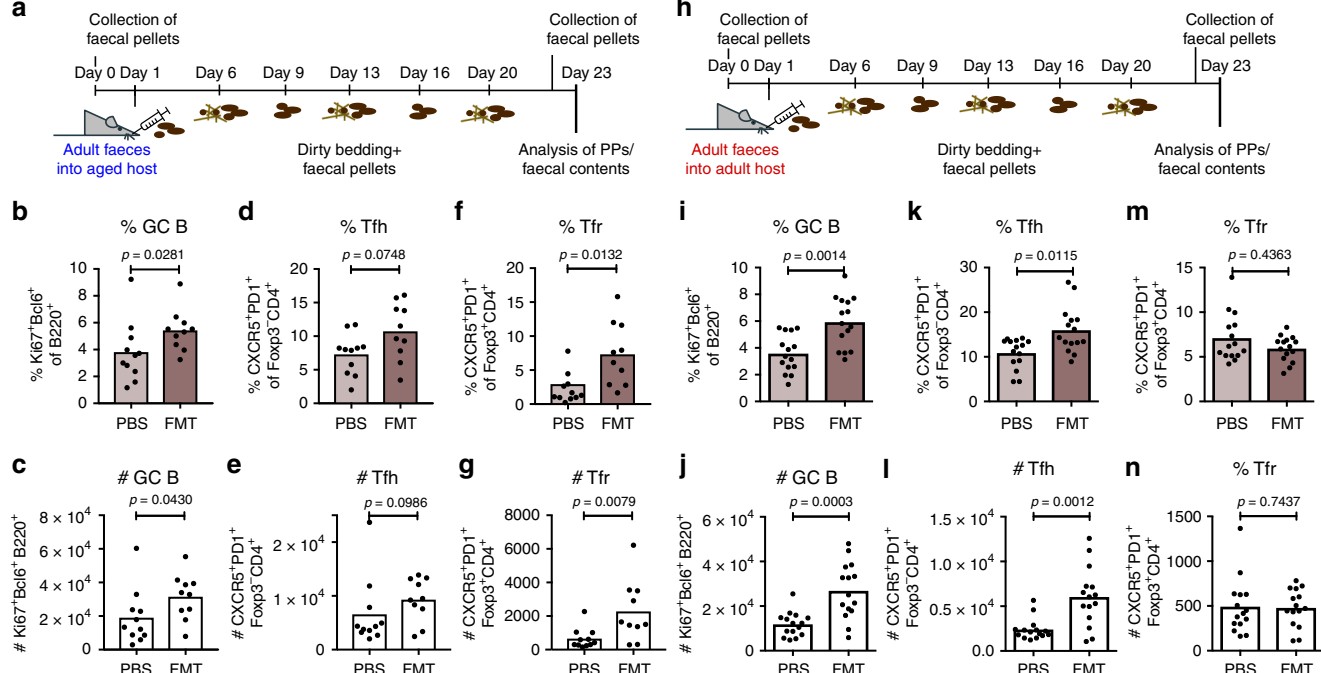

**Fig. 5** Faecal microbiota transplantation boosts gut germinal centres irrespective of age directionality. **a–g** 21-month-old C57BL/6 males were gavaged with faecal pellets from adult, 3-month-old mice. **a** Experimental outline (created by M. Stebegg) of faecal microbiota transplantation (FMT) where aged C57BL/6 males were given a suspension of faecal pellets from adult C57BL/6 donor mice by oral gavage. The cages of recipient mice were supplemented with fresh faecal pellets and dirty bedding from these donor mice once a week. The percentage and number of B220+Ki67+Bcl6+ germinal centre (GC) B cells (**b**, **c**), CD4+Foxp3−CXCR5+PD-1+ Tfh cells (**d**, **e**) and CD4+Foxp3+CXCR5+PD-1+ Tfr cells (**f**, **g**) in Peyer's patches (PPs) as quantified by flow cytometry. **h–n** 3-month-old C57BL/6 males were gavaged with faecal pellets from aged 21-month-old mice. (**h**) Experimental outline of FMT where adult C57BL/6 males were gavaged with faecal pellets from aged mice. The percentage and number of GC B (**i**, **j**), Tfh (**k**, **l**) and Tfr (**m**, **n**) cells in PPs were quantified by flow cytometry. Bar height corresponds to the mean, each circle represents one biological replicate. Bar plots show the combined results of 2 independent experiments with a total of $n = 10$–15 mice per group. P-values were determined using the Mann-Whitney test in GraphPad Prism6. Source data are provided as a Source Data file

was no increase in the percentage or number of GC B cells, Tfh cells or Tfr cells in C57BL/6 mice that received FMT compared with the PBS-treated controls (Fig. 7c–h). This suggests that the GC reaction in PPs responds specifically to heterochronic faecal transplantation and that host genetics might impact the cross-talk between the gut microbiota and the PP GC reaction.

**Cholera toxin boosts the PPs GC reaction in aged mice**. At peripheral sites in the body, a GC reaction is normally induced in response to foreign antigen. To determine whether FMT can boost the mucosal immune response to foreign antigen, we immunised 23-month-old C57BL/6 mice with cholera toxin coupled to the hapten NP (NP-Ctx) by oral gavage three times at weekly intervals, either with or without prior FMT from 3-month-old donors. Assessment of mucosal antibody responses in the faecal contents of the ileum showed that there was no difference in anti-CTx and anti-NP IgA titres between mice that received FMT and PBS controls (Fig. 8a, b). Further, there was no difference in the titre of high-affinity anti-NP2 IgA or in the ratio of NP2/NP20 binding antibodies (Fig. 8c, d), a measure of affinity maturation. Consistent with this, serum anti-CTx and anti-NP IgG1 antibody titres and affinity were not influenced by FMT in aged mice (Fig. 8e–h). Assessment of the GC reaction in the PPs of these animals, surprisingly, showed that immunisation with NP-CTx alone was sufficient to enhance the GC response in 23-month-old animals independently of FMT, with no changes in Tfh or Tfr cell number (Fig. 8i–n). This indicates that FMT, like the potent immunogen CTx, might boost the GC reaction in an

adjuvant-like manner to rescue the diminished GC reaction in the PPs of aged mice.

## Discussion
The composition of the commensal gut microbiota changes with age, which is linked with an increase in age-associated morbidities such as frailty[2,4] and intestinal disorders[30], but the cause of this shift in the gut microbial composition is unknown. Here we sought to determine whether there is a causal link between age-dependent changes in the microbiome and the defective GC reaction in PPs of aged mice. We show that the diminished GC reaction in the PPs of aged mice can be rescued by co-housing with younger animals, as well as by transplantation of faecal microbiota from adult mice. These data demonstrate that the defective germinal centre response in aged mice is not a cell intrinsic feature of the ageing immune system and can be restored by replenishment of the microbiome or stimulation with cholera toxin.

The ageing-related decline in the GC response in LNs has been linked to a decrease in the functions of multiple cell types, including antigen-presenting cells, T and B cells[31]. However, it is not clear whether these age-associated defects are predominantly caused by accumulation of immune cell-intrinsic defects, or reduced stimulation of immune cells via their microenvironment[32]. Our data suggest that, in the PPs of aged mice, the capacity of GC B cells to respond to antigen is not impaired in a cell-intrinsic manner, but can be rescued by stimulation from the microbiota or cholera toxin. Interestingly, the GC reaction in PPs

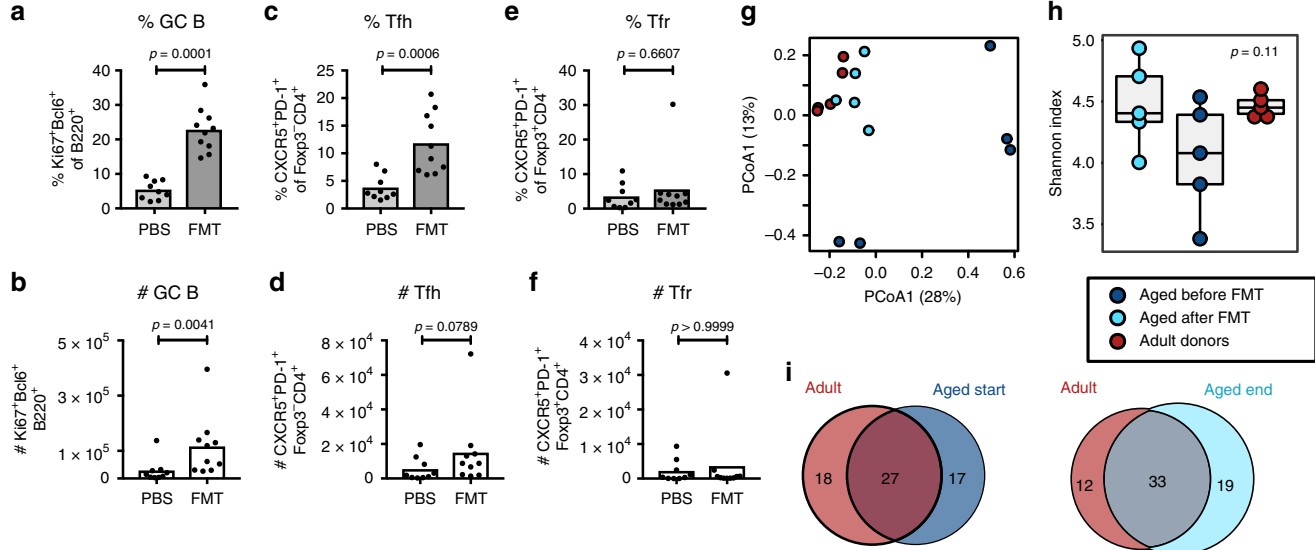

**Fig. 6** Faecal microbiota transplantation boosts the gut germinal centre response of aged BALB/c mice. 21-month-old BALB/c mice were given a suspension of faecal pellets taken from 3-month-old mice by oral gavage. The cages of aged recipients were supplemented with fresh faecal pellets and dirty bedding from these donors once a week. **a–f** Bar plots show the combined results of 2 independent experiments. The percentage and number of B220+Ki67+Bcl6+ germinal centre (GC) B cells (**a**, **b**), CD4+Foxp3-CXCR5+PD-1+ Tfh cells (**c**, **d**) and CD4+Foxp3+CXCR5+PD-1+ Tfr cells (**e**, **f**) in Peyer's patches as quantified by flow cytometry. Bar height corresponds to the mean, each circle represents one biological replicate. Bar plots show the combined results of 2 independent experiments with a total of n = 9–10 mice per group. *P*-values were determined using the Mann–Whitney test in GraphPad Prism6. **g–i** 16S rRNA sequencing data were generated from faecal pellets collected from 5 adult donor mice and 5 aged BALB/c mice before and after faecal microbiota transplantation (FMT). **g** Bray–Curtis PCoA of FMT samples. **h** Shannon diversities with an overall *p*-value generated from ANOVA tests. **i** Venn diagrams showing the numbers of shared and unique bacterial species detected in adult donor mice and aged mice before FMT (start; left) or after FMT (end; right). Source data are provided as a Source Data file

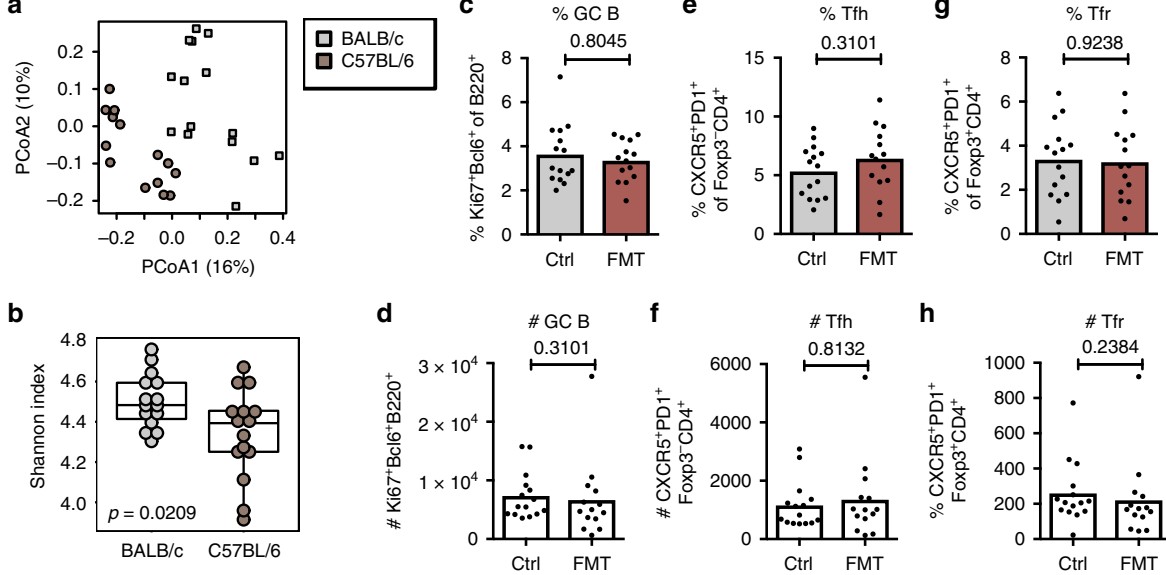

**Fig. 7** Faecal microbiota transplantation between adult mice of different strains does not affect gut germinal centres. **a**, **b** 16S rRNA sequencing data generated from faecal pellets collected from adult 3-month-old C57BL/6 and BALB/c mice. (**a**) Bray-Curtis PCoA comparing samples from 3-month-old BALB/c and C57BL/6 mice. **b** Shannon diversities, with a *p*-value based on ANOVA tests. **c–h** Three-month-old C57BL/6 mice were given a suspension of faecal pellets taken from 3-month-old BALB/c mice by oral gavage to achieve faecal microbiota transplantation (FMT). In addition, the cages of C57BL/6 recipients were supplemented with fresh faecal pellets and dirty bedding from BALB/c mice once a week. Control mice were gavaged with PBS. The percentage and number of B220+Ki67+Bcl6+ GC B cells (**c**, **d**), CD4+Foxp3-CXCR5+PD-1+ Tfh cells (**e**, **f**) and CD4+Foxp3+CXCR5+PD-1+ Tfr cells (**g**, **h**) in Peyer's patches as quantified by flow cytometry. Bar plots show the combined results of two experiments with a total of n = 14–15 mice per group. Bar height corresponds to the mean, each circle represents one biological replicate. *P*-values were determined using the Mann-Whitney test in GraphPad Prism6. Source data are provided as a Source Data file

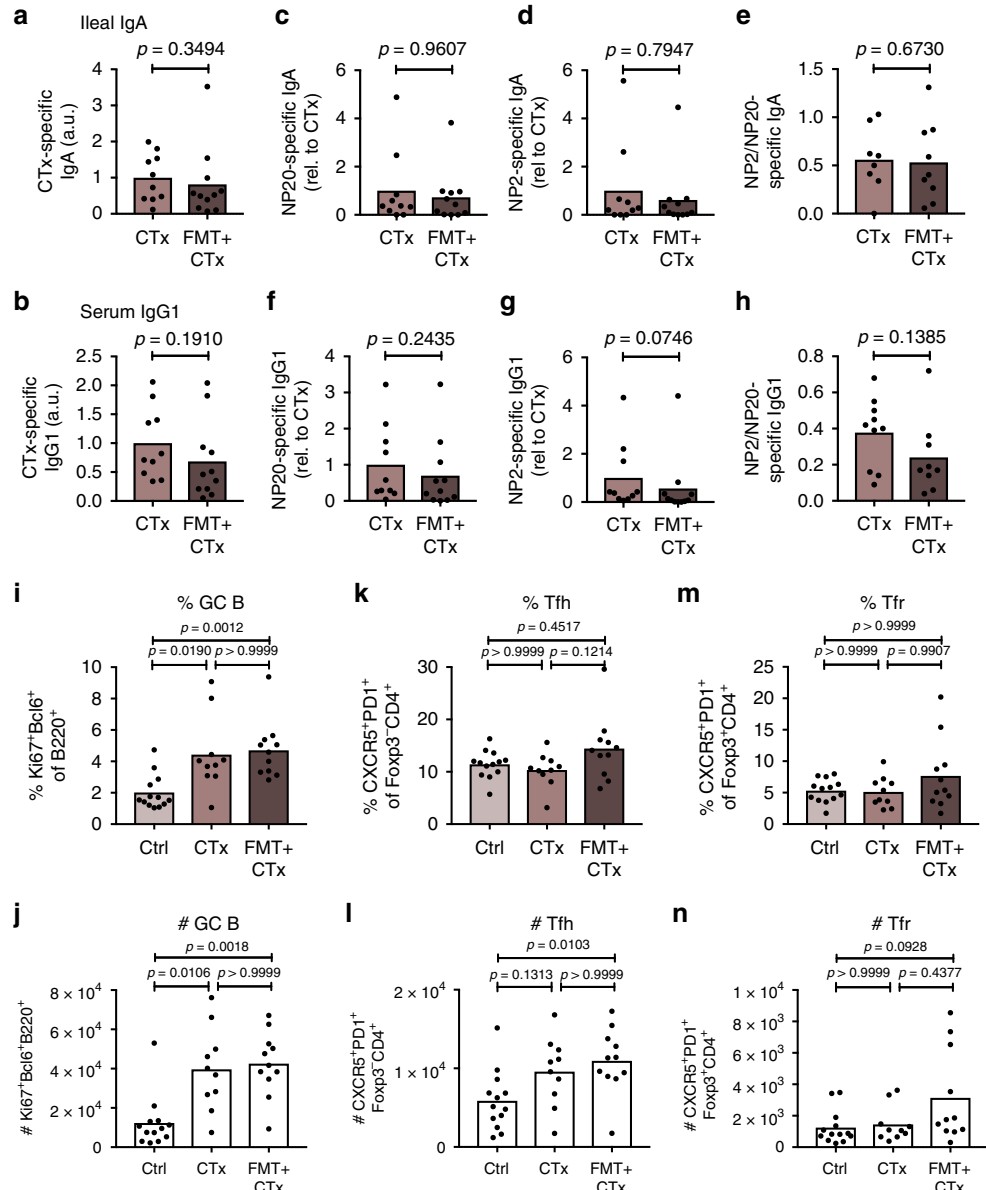

**Fig. 8** Faecal microbiota transplantation does not enhance NP-CTx-specific immune responses in the gut. 21-month-old C57BL/6 mice received faecal microbiota transplantation (FMT) or PBS by oral gavage as described above, followed by three oral immunisations with NP-CTx once a week. After three weeks, NP-CTx-specific antibody levels were analysed in PBS-gavaged, NP-CTx-immunised 23-months-old C57BL/6 mice (CTx) compared with aged mice receiving FMT plus NP-CTx (FMT + CTx). Peyer's patch (PP) germinal centre (GC) responses in these mice were compared with naïve, 23-month-old C57BL/6 mice (Ctrl). **a–d** Antigen-specific IgA levels against CTx (**a**) and NP (**b–d**) in faecal contents from the ileum were assessed by ELISA. NP20-specific IgA (**b**), high-affinity NP2-specific IgA (**c**) and the ratio of NP2-to-NP20-specific IgA antibodies (**d**) were used as a measure for affinity maturation. **e–h** Antigen-specific IgG1 levels against CTx (**e**) and NP (**f–h**) in the serum were assessed by ELISAs for CTx-specific IgG1 (**e**), NP20-specific IgG1 (**f**), high-affinity NP2-specific IgG1 (**g**) and the ratio of NP2-to-NP20-specific IgG1 antibodies (**h**). **i–n** The percentage and number of $B220^+Ki67^+Bcl6^+$ GC B cells (**i**, **j**), $CD4^+Foxp3^-CXCR5^+PD-1^+$ Tfh cells (**k**, **l**) and $CD4^+Foxp3^+CXCR5^+PD-1^+$ Tfr cells (**m**, **n**) in PPs were quantified by flow cytometry. Bar plots show the combined results of 2 independent experiments with a total of $n = 10–13$ mice per group. Bar height corresponds to the mean, and each circle represents one biological replicate. P-values were determined using Mann–Whitney tests (**a–h**) or the Kruskal–Wallis test with Dunn's multiple testing correction (**i–n**) in GraphPad Prism6. Source data are provided as a Source Data file

was more strongly boosted following FMT than by co-housing. If the increase in magnitude of the GC response in aged mice is transient, this discrepancy could be explained by the duration of the different experiments, as we assessed the PPs GC reaction 23-days after FMT, and 30–40 days after the start of cohousing. The observation that not only microbial transfer, but also the potent mucosal antigen, cholera toxin can boost the GC response in older animals indicates that the expansion of the GC reaction by FMT is not due to reactivation of commensal-specific memory B

cells that persist in aged mice after their microbiome changes. Rather it suggests that the use of strong immunogens, or potent adjuvants, can enhance GC responses in older individuals.

Cross-strain FMT from BALB/c into C57BL/6 did not boost the GC reaction. This is surprising, as the differences in gut microbial composition between C57BL/6 and BALB/c mice are similar to the differences observed between adult and aged mice in which FMT does enhance PP GC responses. While we have not been able to dissect the mechanism behind this, host genetics

might play a role in shaping the cross-talk between the gut microbiome and PP GC responses. Consistent with existing literature, we observed clear differences in mucosal IgA levels between BALB/c and C57BL/6 mice (Supplementary Fig. 7F-I)[33]. Fransen et al. linked reduced levels of mucosal IgA in C57BL/6 mice with their impaired ability to mount antigen-specific IgA antibody responses to a non-invasive bacterial species[33]. Thus, the genetic predisposition of C57BL/6 mice for reduced IgA production might impair their ability to mount PP GC responses to BALB/c-derived commensals.

It is well established that the gut microbiome is affected by ageing, but there is no consensus on how exactly the gut microbiome changes with age. This is probably due to the high variability detected in microbiomes from different geographical locations, in both mice and humans[34–36]. Several studies on the human microbiome reported reduced bacterial diversity in aged individuals[4,30] and Claesson et al. observed larger inter-individual variability in older individuals compared with young controls[3]. Our study reports the same general trends—increased inter-individual variability in aged mice of both BALB/c and C57BL/6 mice, and a reduced bacterial diversity in aged BALB/c mice. The differences in the microbiome observed between strains of mice and between the sexes of the same strain, all of which were aged in the same animal facility under the same environmental conditions, indicate that interactions between age, sex and genetics play a role in shaping age-associated alterations of the gut microbiome.

The change in gut microbial composition with age has been linked with many age-associated disorders[7,37]. Transfer of an aged microbiome into germ-free mice causes systemic inflammation[7,38], suggesting that the aged microbiome itself may contribute to so-called inflammaging. Our results indicate that replenishing the microbiome of aged mice with that of a younger animal can boost the local GC reaction, which may have implications for the overall health of the organism. Consistent with this hypothesis, remodelling of the gut microbiome in *Drosophila melanogaster* has been shown to increase lifespan[39,40]. Similarly, middle-aged killifish colonised with a young microbiome were found to live longer than untreated fish[41] and bacterial-derived indoles were shown to increase the lifespan of mice[42]. These data suggest that there is a direct link between the phenotypes associated with ageing and age-associated changes in the gut microbiome. Previous studies showed that supplementation of older humans or mice with prebiotics and probiotics results in changes of gut microbial composition and can improve gut immunity in older individuals[43,44]. Further, the transfer of a young microbiome into aged mice increases protection against *C. difficile* infection[37], indicating that the microbiota of young animals can functionally boost intestinal protection. This makes the gut microbiome a possible target for the treatment of a range of age-associated symptoms. FMT[45], probiotics[44], co-habitation[46] and diet[47] all have an impact on the composition of the gut microbiome and could prove to be innovative interventions to facilitate healthy ageing.

## Methods

**Animals**. Mice were bred and maintained in the Babraham Institute Biological Support Unit. No primary pathogens or additional agents listed in the FELASA recommendations[48] were detected during health monitoring surveys of the stock holding rooms. Ambient temperature was ~19–21 °C and relative humidity 52%. Lighting was provided on a 12 h light: 12 h dark cycle including 15 min 'dawn' and 'dusk' periods of subdued lighting. After weaning, mice were transferred to individually ventilated cages (GM 500: Techniplast) with 1–5 mice per cage. Mice were fed CRM (P) VP diet (Special Diet Services) ad libitum and received seeds (e.g., sunflower, millet) at the time of cage-cleaning as part of their environmental enrichment. All mouse experimentation was approved by the Babraham Institute Animal Welfare and Ethical Review Body. Animal husbandry and experimentation complied with existing European Union and United Kingdom Home Office

legislation and local standards. Adult mice were 10–14 weeks old, and aged C57BL/6 and BALB/c mice were 90–105 weeks old when used for experiments. All experimental mice were housed in the same room. To control for changes in the microbiome due to circadian rhythm, FMT and faecal pellet collections were always performed at 10.00 a.m. and 3.00 p.m., respectively. Due to limited availability of aged male BALB/c mice, all BALB/c experiments were conducted with females. For C57BL/6 mice, experiments were conducted with both male and female mice.

**Co-housing mice**. Two to three adult females were consolidated with 2–3 aged females per cage. After 30–40 days of co-housing, PPs were harvested from all mice for flow cytometric analysis. Faecal pellets were collected from all mice before co-housing and at the end of the experiment. Faecal samples were stored at −80 °C for 16S rRNA sequencing.

**Subcutaneous immunisations**. Co-housed BALB/c mice were immunised with NP-KLH (4-Hydroxy-3-nitrophenylacetyl-Keyhole Limpet Hemocyanin; Biosearch Technologies #N-5060–25) in Imject Alum (Thermo Scientific #77161). NP-KLH was first diluted in PBS, then Alum was added dropwise to the solution while shaking until a final concentration 500 µg/ml NP-KLH was reached. After 30 min of vortexing, 100 µl of the emulsion were injected subcutaneously (s.c.) into the hind flanks of recipient mice. Mice were euthanised on day 14 after immunisation.

**Faecal microbiota transplantation (FMT)**. FMT was achieved by oral gavage of a faecal slurry. Recipient mice had the food removed from the cage for 2 h prior to FMT. The faecal slurry was obtained by pooling faecal pellets from 8–14 donor mice. The pellets were weighed and resuspended by vortexing for 1 min in 1 mL PBS per 300 mg of faeces. After pelleting larger particles by centrifugation at $500 \times g$ for 5 min, the supernatant was collected for FMT. Each recipient mouse received 150 µl of faecal slurry by oral gavage. The remaining slurry was stored at −80 °C for 16S rRNA sequencing. Following FMT, the cages of recipient mice were replenished with dirty bedding and fresh faecal pellets from donor mice once and twice a week, respectively. Faecal pellets for 16S rRNA sequencing were collected the day before FMT and at the end of the experiment, and were stored at −80 °C for DNA extraction. Twenty-three days after FMT, PPs were harvested for flow cytometric analysis.

**Oral immunisations with NP-CTx**. NP-CTx was generated by conjugating NP-e-Aminocaproyl-OSu (NP; Biosearch Technologies #N-1021–100) with cholera toxin (CTx; Sigma #C8052) following a protocol adapted from N. Lycke by A. Iseppon and B. Stockinger[49]. First, 2 mg/ml CTx was dialysed in distilled water for 2 days at 4 °C in a Slide-A-Lyzer Dialysis Cassette (0.5–3 ml size with a 10 K cut-off; Thermo Fisher #66380) before mixing it with an equal volume of 0.1 M NaHCO₃ and 20 equivalents of 10 mg/ml NP-eA-OSu per mole CTx. The mixture was incubated overnight at 4 °C during constant rotation and then dialysed twice against 0.05 M NaHCO₃ followed by PBS. The final protein concentration was determined using a BCA assay (Thermo Fisher #23227).

For oral immunisations mice had the food removed from the cage for 2 h prior to administration of antigen. They were then orally gavaged with 200 µl PBS containing 37.5 µg/ml CTx plus 37.5 µg/ml NP-CTx. When oral immunisations were combined with FMT, oral immunisations with NP-CTx/CTx were conducted on day 2, day 8 and day 15 after FMT. Mice were harvested on day 22 after FMT. Control groups were gavaged with PBS only. On day 22, PPs, mLNs, blood and faecal contents were harvested.

**Flow cytometry of Peyer's patches and lymph nodes**. A single cell suspension from dissected Peyer's patches (PPs) and mesenteric or draining lymph nodes (LNs) was generated by pressing the tissues through a 70 µm mesh in 2% foetal bovine serum in PBS. Cell numbers and viability were determined using a CASY TT Cell Counter (Roche). Antibody stainings of $1–3 \times 10^6$ cells were performed in 96 well U-bottom plates. Cells were first stained with LIVE/DEAD® Fixable Blue Dead Cell Stain (Invitrogen #L23105; diluted 1:1000 in PBS) and then incubated with FcR block for 15 min (anti-mouse CD16/32, clone 93, eBioscience). Surface antibody staining was performed for 60 min at 4 °C in Brilliant Stain Buffer (BD Biosciences #563794). For intranuclear staining, cells were fixed with the eBioscience Foxp3/Transcription Factor Staining Buffer (#00-5323-00) for 30–60 min. Staining with anti-Foxp3, anti-Ki67 and anti-Bcl6 antibodies were performed for 60 min at 4 °C in 1× Permeabilization buffer (eBioscience #00-8333-56). Samples were acquired on a LSRFortessa (BD Biosciences) with stained UltraComp eBeads™ Compensation Beads (Invitrogen #01-2222-41) as compensation controls. Flow cytometry data were analysed using FlowJo v10 software (Tree Star). The antibodies used are listed in Table 1.

**Flow cytometry of IgA-coating of faecal bacteria**. To assess IgA-coating of faecal bacteria, we adapted a protocol described by Sidonia Fagarasan and Mikako Maruya[19]. Briefly, the faecal contents of the ileum and colon were collected. The faecal contents were weighed and incubated for 10 min on ice in 12 µl sterile-filtered PBS per mg faeces. The samples were then vortexed at full speed for 1 min

**Table 1 List of antibodies used for flow cytometry**

| Antibody | Supplier (Clone) | Dilution |
|---|---|---|
| PE/PE-Cy7-coupled anti-mouse Bcl6 | BD Biosciences (K112-91) | 1:100 |
| APC-AF780-coupled anti-mouse PD1 | eBioscience (J43) | 1:200 |
| APC/Foxp3-coupled anti-mouse Foxp3 | eBioscience (FJK-16S) | 1:100–1:200 |
| AF488/AF700-coupled anti-mouse Ki67 | eBioscience (SolA15) | 1:100 |
| BV421-coupled anti-mouse CXCR5 | Biolegend (L138D7) | 1:100 |
| V500/PE/BV605-coupled anti-mouse CD4 | Biolegend (RM4-5) | 1:400–1:800 |
| BV510/BV785-coupled anti-mouse B220 | Biolegend (RA3-6B2) | 1:200–1:400 |
| PerCp-Cy5.5-coupled anti-mouse CD44 | Biolegend (IM7) | 1:200 |
| AF674/FITC-coupled anti-mouse IgA | Southern Biotech (1040-02/-31) | 1:100 |
| APC-Cy-7-coupled anti-mouse IgK | BD Biosciences (187.1) | 1:100 |

and spun at $500 \times g$ at 4 °C for 5 min to pellet bigger particles. The supernatant containing faecal bacteria was transferred to a fresh tube and spun at $10,000 \times g$ at 4 °C for 5 min to pellet bacteria. Bacteria were blocked in 2% bovine serum albumin (BSA) in PBS for 15 min, followed by staining with anti-IgA (Southern Biotech #1040–02) and anti-IgK (BD Biosciences, Clone 187.1) antibodies in 2%BSA/PBS for 45 min on ice. Stained bacteria were fixed in 4% PFA overnight at 4 °C. The next day, the cells were stained with DAPI (Invitrogen #D1306; 1:1000 in a staining buffer containing 0.01% Tween and 1 mM EDTA) and AF594-coupled wheat germ agglutinin (Invitrogen #W11262; 1:100 in 3 M KCl solution) to distinguish between gram-positive and gram-negative bacteria. The samples were acquired at 5000 events/s on a Fortessa 5 (BD Biosciences) with the SSC threshold set to 200. Single stains served as compensation controls. The antibodies used are listed in Table 1.

**ELISA sample preparations.** At indicated time points, blood or faecal contents were obtained from mice. Blood samples were obtained by cardiac puncture and spun at 13,000 rpm for 15 min at room temperature. The serum supernatants were collected and stored at −20 °C for ELISAs.

In oral immunisation experiments, faecal contents from the colon and ileum were weighed and incubated for 10 min on ice in 5 ml sterile-filtered PBS per gram of faeces (200 mg/ml faecal input). The samples were then vortexed at full speed for 1 min and spun at $10,000 \times g$ at 4 °C for 10 min to pellet bacteria and debris. The supernatant containing free IgA was harvested and stored at −20 °C. In FMT experiments where IgA-coating of faecal bacteria was assessed, faecal contents from the colon and ileum were incubated for 10 min on ice in 12–14 ml sterile-filtered PBS per g of faeces (70–85 mg/ml faecal input). The samples were then vortexed at full speed for 1 min and spun at 500 g at 4 °C for 5 min to pellet bigger particles. 80–100 µl of the supernatant containing faecal bacteria was transferred to a fresh tube, topped up with PBS to reach a total volume of 1 ml and spun at $10,000 \times g$ at 4 °C for 5 min to pellet bacteria. The supernatant containing free IgA (at a final concentration of 5–8.5 mg/ml faecal input) was harvested and stored at −20 °C, while the bacterial pellet was resuspended in blocking buffer and stained for flow cytometry as described above.

**Quantitation of antigen-specific antibodies.** ELISA plates (Thermo Scientific 96 F Maxisorp #456537) were coated overnight at 4 °C with 10 µg/ml NP20-BSA (Biosearch Technologies #N-5050H-100), 2.5 µg/ml NP7-BSA (Biosearch Technologies #N-5050L-100), 2.5 µg/ml NP2-BSA (Biosearch Technologies #N-5050L-100) or 1 µg/ml CT in PBS. The next day, plates were washed 4× in 0.05% Tween 20 wash buffer and blocked with 1% BSA in PBS for 1 h at room temperature. Starting dilutions of ileal supernatants were 10–20 mg/ml faecal input (1:5–10 dilution of frozen supernatants) in 1% BSA/PBS. This initial dilution was titrated down the plate at a 1:4 ratio. The plates were incubated for 2 h at room temperature and after another wash step the plates were incubated with 50 µl of polyclonal goat anti-mouse IgG1 HRP-conjugated antibodies (Abcam #ab97240; 1:10,000 in PBS) or polyclonal goat anti-mouse IgA HRP-conjugated antibodies (Bethyl #A90–103P; 1:25,000 in PBS) for 2 h at room temperature. The plates were developed with 100 µl/well TMB (Biolegend #421101) for upto 20 min, when the reaction was stopped with 50 µl/well 0.5 M H2SO4. A PHERAstar FS microplate reader (BMG Labtech) was used to measure absorption at 450 nm. Absorbance values from serially diluted samples were plotted and values which fell into the linear range of the curve were selected to calculate endpoint titres.

**Quantitation of faecal IgA.** Quantitation of faecal IgA was performed using Bethyl's Mouse IgA ELISA Quantitation Set (#E90–103). ELISA plates were coated overnight at 4 °C with 1 µg/ml goat anti-mouse IgA, then blocked with 1% BSA in PBS for 1 h at room temperature. Faecal supernatants were loaded onto ELISA plates as duplicates in starting concentrations of 0.1–0.5 mg/ml for ileal supernatants and 5–8.5 mg/ml for colonic supernatants. Plate-bound IgA was detected using 50 µl of Bethyl's polyclonal goat anti-mouse IgA HRP-conjugated antibodies (1:30,000 in PBS) for 2 h at room temperature. Plates were developed with 100 µl/well TMB for upto 20 min, when the reaction was stopped with 50 µl/well 0.5 M H2SO and absorption was measured at 450 nm using a PHERAstar FS microplate

reader (BMG Labtech). Bethyl's standard dilution of mouse reference serum was used for the quantitation of free IgA in faecal supernatants.

**Bacterial 16S rRNA sequencing: DNA isolation.** Bacterial DNA isolation from stored faecal matter was performed using the QIAamp PowerFecal DNA Kit (Qiagen #12830-50) following the manufacturer's instructions. First, samples were homogenised by bead beating using a FastPrep24 machine at 5 m/s for 50 s. Total genomic DNA was captured on a silica membrane and the QIAmp Inhibitor Removal Technology was used to remove common substances that can interfere with downstream applications. In the final step, purified DNA was eluted in 70 µl of Solution C6. DNA concentrations were determined using the Qubit dsDNA High Sensitivity Assay Kit (Invitrogen #Q32854) on a Qubit 4 Fluorometer (Invitrogen).

**16S rRNA sequencing library preparation and sequencing.** High-throughput sequencing of the V3-V4 hypervariable region of the bacterial 16S rRNA gene was performed by the Beijing Genomics Institute (BGI) where, first, DNA integrity was assessed by agarose gel electrophoresis. After DNA quantification by Qubit Fluorometer, 30 ng DNA per sample were utilised to PCR-amplify the V3-V4 region using 341 F forward (ACTCCTACGGGAGGCAGCAG) and 806 R reverse fusion primers (GGACTACHVGGGTWTCTAAT). The PCR products were purified using Agencourt AMPure XP beads (Beckman Coulter) for library validation using the Agilent 2100 bioanalyzer instrument (Agilent DNA 1000 Reagents). All libraries passed the quality control and were sequenced on an Illumina MiSeq Platform using the MiSeq reagent kit (2 × 300 bp paired-end reads, Illumina). The resulting sequencing reads were supplied by BGI as demultiplexed fasta files for downstream analysis.

**Bacterial 16S rRNA sequencing: data analysis.** 16S rRNA sequencing analysis was performed using QIIME2 (www.qiime2.org[50]). Successfully merged reads were quality filtered in QIIME2 using default settings. Sequences were clustered into operational taxonomic units (OTUs) based on similarity to the annotated bacterial sequences provided by Silva (v132 SSU; https://www.arb-silva.de/[51]; 99% sequence similarity cut-off). For this, classifiers were trained in QIIME2 based on the Silva database and the V3-V4 primers used for library preparation. Taxa bar plots were generated using QIIME2. Statistical analysis was performed with the Calypso software (v8.82; cgenome.net/calypso/[52]) applying default parameters. Cumulative-sum scaling (CSS) was applied, followed by $\log_2$ transformation to account for the non-normal distribution of taxonomic counts data. Bray-Curtis PCoA was performed based on OTUs. Differences in bacterial diversity (Shannon index) between study groups were evaluated using ANOVA. Calypso was also used to generate KRONA plots[53]. Bacterial families whose abundance was statistically significantly different between groups were determined by ANOVA, using false discovery rate (FDR)-adjusted $p$-values ≤ 0.05 as the cut-off. Venn Diagrams were generated using the Metachart App (https://www.meta-chart.com/venn#/display).

**Statistics.** All experiments were repeated 2–6 times with 4–10 mice per group each. Differences between experimental groups, as determined by flow cytometry, were assessed within the Prism 6 software (GraphPad) using the two-tailed non-parametric Mann–Whitney test or Kruskal–Wallis tests with Dunn's multiple testing correction when more than two groups were compared. $P$-values ≤ 0.05 were considered statistically different. Statistical differences between bacterial families based on 16S-seq data were determined by ANOVA, using FDR-adjusted $p$-values ≤ 0.05 as the cut-off.

**Reporting summary.** Further information on research design is available in the Nature Research Reporting Summary linked to this article.

## Data availability

The 16S rDNA sequencing data generated for this study have been deposited at the European nucleotide archive (ENA); study code PRJEB31652, accession number ERP114231. Source data underlying all figures except Supplementary Fig. 1 are provided with the paper as a Source Data File. Further data in support of our findings are available from the corresponding author upon request.

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

## Acknowledgements

We are grateful to Dr Geoff Butcher, Dr Dario Valenzano and Jens Seidel for feedback on this manuscript. We acknowledge the contribution of the Babraham Institute Biological Support Unit staff, who performed in vivo treatments of our animals and took care of animal husbandry. We thank the staff of the Babraham Flow Cytometry Facility for their technical support. The authors are grateful to Jonathan Clark and Melanie Stammers for their input on the experiments with ageing animals. This study was supported by funding from the Biotechnology and Biological Sciences Research Council (BBS/E/B/000C0407, BBS/E/B/000C0427 and the Campus Capability Core Grant to the Babraham Institute), the European Research Council (637801 TWILIGHT), and the European Union's

Horizon 2020 research and innovation programme "ENLIGHT-TEN" under the Marie Sklodowska-Curie grant agreement No.: 675395. T.P.J. is recipient of a PhD scholarship from the Biotechnology and Biological Sciences Research Council (BBSRC) of the United Kingdom. Research in the C.C. laboratory is funded by grants from the Isaac Newton Trust, the Isaac Newton Trust/Wellcome Trust/University of Cambridge joint research grant scheme and by the Royal Society (UK).

## Author contributions

M.S. and M.A.L. planned the experiments and wrote the manuscript. M.S., A.S.C. and S.I. conducted experiments and M.S. analysed the data. T.P.J. and C.C. assisted with the 16S-seq analysis, while C.G. supplied preliminary 16S-seq data. M.A.L. conceived the study, obtained funding and supervised the study. All authors read and approved the final version of the manuscript.

## Additional information

**Competing interests:** The authors declare no competing interests.

