## [Peer Review File · Nature Communications]

Reviewers' comments:

Reviewer #1 (Remarks to the Author):

The manuscript "Heterochronic faecal transplantation rejuvenates the gut germinal center reaction" submitted by Dr. Linterman and her colleagues addresses a very intriguing question about the source of age-related changes in the immune system. They propose that since immune system changes (such as the reduced number of germinal center B cells in Peyer's Patches) and changes in the gut microbiome both occur with aging, these two factors could be related. In fact, their results very nicely show that by transplanting a young gut microbiome into an older mouse, the frequency and number of germinal center B cells in the Peyer's Patches becomes equal to that seen in younger mice. This study is scientifically novel and this is an important topic. Furthermore, the experiments are well done and the manuscript is well written. There are some points that need to be addressed by the authors:

1. The authors show that there is a reduction in the number of germinal center B cells in Peyer's Patches of older mice when compared to young, but that this is not the case in the mesenteric lymph nodes (Supplemental Figure 1). They need to comment further on this finding and how it relates to their hypothesis of the gut microbiome impacting immunity. Did they examine any other lymph nodes during their studies and were these impacted by microbiome transplantation? What is the overall relevance of the gut microbiome to systemic immune responses?

2. While they show that after heterochronic faecal transplantation, young and aged mice now have similar frequencies and numbers of germinal center B cells in their Peyer's Patches, there is no demonstration that this has an impact on any immune response. Is the level of antibody in the gut changed? Is there an improved response following an infection? Without this information, this is simply an intriguing observation.

3. The authors do not adequately discuss the results of the experiment that they show in Figure 5 H-N, where they transplant aged faeces into younger hosts and observe an enhancement of germinal center B cells and T follicular helper cells in the Peyer's Patches. This is a puzzling result and seems to go against their main hypothesis that the gut microbiome in the older mice is immunosuppressive.

Other issues to address:

In the bottom row of graphs in Figure 5, the % should be #.

Reviewer #2 (Remarks to the Author):

This ms deals with the very timely issue of the effects of the gut microbiota on aspects of intestinal immunity, in particular on germinal center (GC) reactivity. The work is technically well done and the data presented clearly.

The authors initially demonstrated an age-associated shift of the microbiota profile. They then moved onto showing that microbiota transfer (MT) from adult to ageing mice improved GC reactivity. Parallel experiments however, also showed that the effect of MT from aged donors into young/adult recipients surprisingly had a similar effect, thus showing that the effects on GC reactivity is independent of the age of the donor and its composition. Somehow this seems to argue against a direct role of microbiota on GC reactivity. Equally intriguing is the finding that cross-species MT does not affect GC reactivity at all. In my opinion these are the most important observations reported here; however, the authors fell short of discussing the potential implications thoroughly.

The authors also state that this finding can help to design novel microbe-based approach to sort out age-related disturbances (possibly of immune nature?). However, I do not see how. Do they envisage that transferring microbiota from young or ageing donors (they demonstrated that the effect of MT is

independent of the donor's age) to ageing recipients may improve their intestinal immunity? The data do not support this conclusion; indeed, it would appear from their data that MT from individuals not genetically-matched would not work. I think that these results should be given the appropriate attention.

Also, in order to claim that MT can help to restore a young-like immune system the authors should also provide data on the effect of MT on the business end of the immune system. Several reports have shown that ageing is characterized by a sharp decline of IgA-mediated response. I suggest to evaluate mucosal (and systemic) Ab response to orally administrated antigen in post-FMT mice. Alternatively, they should provide at least data on lamina propria IgA-producing plasma cells and IgA levels in intestinal washings.

Reviewer #3 (Remarks to the Author):

The authors have demonstrated that the microbiome can be transferred between young and old mice and that this influences age-related changes in gut immunity, specifically the immune cell populations of the Peyer's patches including germinal centre B cells, T follicular helper and regulatory cells. These experiments are important because it has been shown that age-related changes in the gut microbiome and intestinal architecture have profound effects on the health of the host. The inclusion of two different strains of mice, which are known to have differences in aging trajectory, mucosal immunity and IgA responses is a particular strength. I am very supportive of this manuscript as it is well written, novel and exciting and there are many important findings in here which I think will be of broad interest to readers of this journal but I have listed two suggestions below that I think would increase the level of mechanistic understanding that the microbiota has on the aging immune system. Major points

1) The findings that the microbiome alter GC B, Tfh, Tfr numbers is really exciting and I expect that this has broad implications for infections and homeostatic interactions with gut microbes. One experiment I would really like to see is whether these changes in cell numbers alter germinal centre function, perhaps in the context of an infectious disease challenge (Cdiff being of particular relevance to the elderly?) or even with regard to the IgA repertoire. As the authors state in the introduction, since IgA shapes the microbiome, could age-related changes in the GC be cause or consequence of dysbiosis? For example, for species lost during the course of aging (and restored by microbiome transfer), is there a loss of IgA that is restored during the microbiome transfer (and could this be detected by a bacterial flow cytometry experiment?). Presumably measures of IgA could be taken in a non-invasive manner so that they could be measured before and after transfer.

2) There is a lot of data in the heterochronic co-housing to think through. For example, the aged Balb/c mice appear to have GC B cells restored (no effect on Tfr) but enhanced Tfh whereas the aged C57bl/6 mice may or may not changes in GC B cells (depending on whether you look at % or #), no effect on age related increases in Tfh and maintain the higher Tfr with age but clearly do have altered microbiota. The experiments in Fig 5 add complexity to this as they imply that any microbiome transfer can increase proportion of all three cell types irrespective of the age of the host. I wonder if the authors have considered whether the (apparent) discrepancy btw other experiments is due to the difference in measuring at 3 wks or 40 days? (ie. Could some of these increased cell numbers be transient). The control here would be the young-young mice or old-old mice to disentangle age of the host versus the process of co-housing (especially since the data in Fig 7 demonstrate that in young mice with an intact microbiota, there are no changes, increasing the evidence that this is microbiome independent).

3) In understanding the functional consequences of age related changes in the GC, I'm reminded of this paper <https://doi.org/10.1016/j.immuni.2015.08.011> describing strain specific differences in IgA, independent of the microbiome. Is it possible that the Balb/c strain is more susceptible to age-related changes in GC? Would this be apparent in a challenge model or in IgA load in the serum or fecal

pellets?

Minor points

1) typo in the introduction 'Those GC B cell which able to ...' = "Those GC B cell which*are* able to ..."

2) I know that taxa plots are a bit pedestrian but In Fig 2G & H I found that there were a few gems that are hard to discern in the Krona plots. For example, the Bacteroidales/Firmicutes rations have been shown to change with age before in both mice and humans (and it is very valuable to the community to demonstrate this is a reproducible change!) and I think that this is much more exaggerated in the Balb/c is interesting (could the Firmicutes in the C57 be coloured green as well for ease of comparison?). I'm also confused about 'uncultured' as a classification – were these 'unclassified' perhaps (I doubt the authors cultured all the strains). Beside the 3% Roseburia in the 3mo BalbC there seems to be an unlabelled group? Some of these gems might be more obvious is taxa plots representing order, genus etc.

Best of luck with an interesting manuscript.

Reviewer #1 (Remarks to the Author):

The manuscript "Heterochronic faecal transplantation rejuvenates the gut germinal center reaction" submitted by Dr. Linterman and her colleagues addresses a very intriguing question about the source of age-related changes in the immune system. They propose that since immune system changes (such as the reduced number of germinal center B cells in Peyer's Patches) and changes in the gut microbiome both occur with aging, these two factors could be related. In fact, their results very nicely show that by transplanting a young gut microbiome into an older mouse, the frequency and number of germinal center B cells in the Peyer's Patches becomes equal to that seen in younger mice. This study is scientifically novel and this is an important topic. Furthermore, the experiments are well done and the manuscript is well written. There are some points that need to be addressed by the authors:

We appreciate the supportive comments about the importance and novelty of this work. The individual points raised by the reviewers are addressed below.

1. The authors show that there is a reduction in the number of germinal center B cells in Peyer's Patches of older mice when compared to young, but that this is not the case in the mesenteric lymph nodes (Supplemental Figure 1). They need to comment further on this finding and how it relates to their hypothesis of the gut microbiome impacting immunity. Did they examine any other lymph nodes during their studies and were these impacted by microbiome transplantation? What is the overall relevance of the gut microbiome to systemic immune responses?

To address the contribution of age and microbiome transplantation to systemic immune responses we assessed the cellular and humoral response to the model antigen NP-KLH in Alum after subcutaneous immunization, either in the context of cohousing young and aged mice, or in mice of different ages housed separately. These data have been included as Supplementary figure 6, and are referred to in the manuscript on page 6. The new text and figure are below for ease of reading:

"To determine whether co-housing impacts systemic immune responses we assessed the response to subcutaneous NP-KLH immunisation in the draining inguinal LNs in 22-month-old BALB/c mice that were either co-housed with 3-month-old adult mice or with their age-matched littermates. 22-month-old BALB/c mice had a reduced frequency and number of GC B cells in their draining LNs 14 days after immunisation compared to adult mice, irrespective of whether they were cohoused with littermates or with younger animals (Supplementary Figure 6A-C). Total Tfh cell numbers in the inguinal LNs were not influenced by age or cohousing after immunisation (Supplementary Figure 6D-E). The reduced GC response in aged mice corresponded with decreased antigen-specific antibody production after vaccination and a defect in affinity maturation. This diminished antibody response was not influenced by cohousing (Supplementary Figure 6G-I). Together this indicates that systemic immune responses are not influenced by the microbial transfer that occurs during co-housing, and suggests that the impact of microbial transfer on the GC reaction is limited to the gut."

Because of these findings we have removed the sentences in the introduction and discussion that suggest FMT could be used to boost vaccine responses.

Supplementary Figure 6: Systemic immune responses are not affected by co-housing. Peripheral immune responses were analysed and compared between separately housed and co-housed adult (3-month-old; 3mo) and aged (22-month-old; 22mo) BALB/c mice 14 days after subcutaneous immunisation with NP-KLH/Alum. **(A)** Representative flow cytometric plots of B220⁺Ki67⁺Bcl6⁺ germinal centre (GC) B cells in the draining lymph nodes of NP-KLH-immunised mice. **(B, C)** Quantitation of B220⁺Ki67⁺Bcl6⁺ GC B cells in percentage **(B)** and cell numbers **(C)**. **(D)** Representative flow cytometric plots of CD4⁺CXCR5⁺PD-1⁺ Tfh cells in the draining lymph nodes. **(E, F)** Quantitation of CD4⁺CXCR5⁺PD-1⁺ Tfh cells in percentage **(E)** and cell numbers **(F)**. **(G-I)** Affinity maturation of NP-specific IgG1 antibodies in the same mice as measured by ELISAs of NP20-specific IgG1 **(G)**, high-affinity NP7-specific IgG1 **(H)** and the ratio of NP7-to-NP20-specific IgG1 antibodies **(I)**. Bar plots show the combined results of two independent experiments with a total of n=13-19 mice per group. Bar height corresponds to the mean, and each circle represents one biological replicate. P-values were determined using the Kruskal-Wallis test with Dunn's multiple testing correction in GraphPad Prism6.

2. While they show that after heterochronic faecal transplantation, young and aged mice now have similar frequencies and numbers of germinal center B cells in their Peyer's Patches, there is no demonstration that this has an impact on any immune response. Is the level of antibody in the gut changed? Is there an improved response following an infection? Without this information, this is simply an intriguing observation.

1. Antibody levels in the gut

To determine whether microbial transfer influences immune responses or antibody responses in the gut, we first compared IgA coating of gut bacteria between young and aged mice. We did not observe an age-dependent change in the frequency of IgA-coated bacteria in either C57BL/6 or BALB/c mice. These data have been included in the

manuscript as Supplementary Figure 4 and described in the text on page 6. Consistent with this there was no age-dependent change in IgA coating after cohousing or FMT, these data have been included in the manuscript as Supplementary Figures 5 and 7 and described in the text on page 6 and 9, respectively. This indicates that the microbiota has a bigger impact on the GC than vice versa. The revised text and figures are below:

“To determine whether there is a link between the poor GC reaction in the PPs of aged mice and antibody regulation of the microbiome, we examined IgA coating of gut bacteria in aged mice. In both the ileum and colon the frequency of IgA coated bacteria was comparable between 3-month-old adult and 22-month-old aged mice from both the C57BL/6 and BALB/c strains (Supplementary Figure 4). This could be explained by previous studies which described normal levels of intestinal IgA and commensal IgA coating in GC-deficient mice, suggesting that commensal-specific IgA can be produced in a GC-independent manner^{27–29}. Taken together, this indicates that the impaired GC reaction in the PPs of aged mice does not influence commensal-reactive IgA, and prompts the hypothesis that the composition of the microbiome may influence the magnitude of the GC response.”

Page 6 in the manuscript:

Supplementary Figure 4: IgA-coating of faecal bacteria is not affected by ageing. Bacterial IgA-coating was assessed in adult (3-month-old; 3mo) and aged (22-month-old; 22mo) mice. **(A)** Gating strategy for IgA-coated bacteria. **(B, C)** Representative flow cytometric plots of IgA-coated faecal bacteria isolated from the ileum **(B)** and colon **(C)** of BALB/c mice. **(D-G)** Quantitation of IgA-coating of bacteria in faecal contents isolated from the ileum **(D, E)** and colon **(F, G)** of adult and aged BALB/c **(D, F)** and C57BL/6 **(E, G)** mice. Bar plots show the combined results of 2-4 independent experiments with a total of n=11-16 mice per group. Bar height corresponds to the mean, and each circle represents one biological replicate. P-values were determined using the Mann-Whitney test in GraphPad Prism6.

“To our surprise, the age-associated reduction of GC B cells in PPs of BALB/c mice was lost upon such co-housing (Figure 3A, B). This correction of the GC reaction was accompanied by an increase in Tfh cells but Tfr cell numbers and IgA coating of commensals were unchanged (Figure 3C-F; Supplementary Figure 5).”

Supplementary Figure 5: Co-housing does not affect IgA-coating of faecal bacteria. Bacterial IgA-coating was assessed in adult (3-4-month-old; 3mo/4mo) and aged (22-month-old; 22mo) mice after co-housing for 40 days. **(A-D)** Quantitation of IgA-coating of bacteria in faecal contents isolated from the ileum **(A, B)** and colon **(C, D)** of cohoused BALB/c **(A, C)** and C57BL/6 **(B, D)** mice. Bar plots show the combined results of 2-4 independent experiments with a total of n=19-30 mice per group. Bar height corresponds to the mean, and each circle represents one biological replicate. P-values were determined using the Mann-Whitney test in GraphPad Prism6.

Page 9 in the manuscript:

“To determine whether FMT can rescue the PP GC reaction in aged mice of another strain, we performed FMT experiments in 22-month-old BALB/c mice (experimental set-up as in Figure 5A). GC B and Tfh cells had significantly expanded in 22-month-old mice 23 days after FMT compared to PBS-treated control mice (Figure 6A-D), but Tfr cells were not changed by FMT treatment (Figure 6E, F). FMT did not affect the levels of PP-resident IgA⁺ B cells, free intestinal IgA or bacterial IgA coating in either mouse strain (Supplementary Figure 7).”

Supplementary Figure 7: Faecal microbiota transplantation (FMT) does not affect gut IgA responses. 21-month-old mice were given a suspension of faecal pellets taken from 3-month-old mice by oral gavage. The cages of these aged recipients were supplemented with fresh faecal pellets and dirty bedding from these donors once a week. A control group received PBS by oral gavage. After 3 weeks, intestinal IgA levels were assessed in adult donor mice (4mo), aged control mice (23mo+PBS) and aged mice receiving FMT (23mo+FMT). **(A)** Representative flow cytometric plots for IgA⁺ B cells (B220⁺IgA⁺) cells in the Peyer's patches (PPs) of adult (3-month-old; 3mo) and aged (22-month-old; 22mo) BALB/c mice. **(B-E)** Quantitation of IgA⁺ B cells in percentage **(B, D)** and cell numbers **(C, E)** in BALB/c **(B, C)** and C57BL/6 **(D, E)** mice by flow cytometry. **(F-I)** Quantitation of faecal IgA in faecal contents isolated from the ileum **(F, H)** and colon **(G, I)** of BALB/c **(F, G)** and C57BL/6 **(H, I)** mice by ELISA. **(J-M)** Quantitation of IgA-coating of bacteria in faecal contents isolated from the ileum **(J, L)** and colon **(K, M)** of BALB/c **(J, K)** and C57BL/6 **(L, M)** mice. Bar plots show the combined results of 2 independent experiments with a total of n=8-16 mice per group. Bar height corresponds to the mean, and each dot represents one biological replicate. P-values were determined using the Kruskal-Wallis test with Dunn's multiple testing correction in GraphPad Prism6.

2. Does FMT improve responses to infection?

It has previously been shown that transfer of faecal bacteria from adult into aged mice increases protection against *C. difficile* infection. This is included in the last paragraph of the discussion of this manuscript to highlight the importance of this treatment to improving health outcomes in older individuals.

Reference 33: Shin, J. H. *et al.* Innate Immune Response and Outcome of Clostridium difficile Infection Are Dependent on Faecal Bacterial Composition in the Aged Host. *J. Infect. Dis.* **217**, 188–197 (2018)

The last paragraph of the discussion reads:

"The transfer of a young microbiome into aged mice increases protection against *C. difficile* infection ³³, indicating that the microbiota of young animals can functionally

boost intestinal immune protection. This makes the gut microbiome a possible target for the treatment of a range of age-associated symptoms. FMT ⁴¹, probiotics ⁴⁰, co-habitation ⁴² and diet ⁴³ all have an impact on the composition of the gut microbiome and could prove to be innovative interventions to facilitate healthy ageing.”

3. The authors do not adequately discuss the results of the experiment that they show in Figure 5 H-N, where they transplant aged faeces into younger hosts and observe an enhancement of germinal center B cells and T follicular helper cells in the Peyer’s Patches. This is a puzzling result and seems to go against their main hypothesis that the gut microbiome in the older mice is immunosuppressive.

While some studies report that the microbiome from aged mice is immune suppressive (Refs 7, 34), we performed the experiment described in Figure 5H-N to test whether FMT specifically boosts the GC response in aged mice or whether it can occur in young mice also, rather than to test the hypothesis that the gut microbiome from older mice is immunosuppressive. We have changed the text on page 8 to make this clearer. The text is below:

“To determine whether the boost in the GC reaction by FMT is exclusive to aged mice, 3-month-old C57BL/6 mice were gavaged with faecal pellets from 22-month-old adult mice (Figure 5H). For this experiment we used C57BL/6 mice because we had evidence of reciprocal microbial transfer between adult and aged mice of this strain during co-housing: this result indicated that the gut microbiome of both adult and aged C57BL/6 mice is receptive to microbial transfer and presented us with a tool to assess whether the transfer of a new microbiome generally enhances the GC reaction irrespective of age, or if it constitutes a unique feature in aged mice. FMT of younger adult mice with faecal pellets from aged mice led to an increase in GC B as well as Tfh cells, while Tfr cell numbers were not affected (Figure 5I-N).”

Other issues to address:

In the bottom row of graphs in Figure 5, the % should be #.
Thank you for spotting this error, it has been corrected.

Reviewer #2 (Remarks to the Author):

This ms deals with the very timely issue of the effects of the gut microbiota on aspects of intestinal immunity, in particular on germinal center (GC) reactivity. The work is technically well done and the data presented clearly.

We appreciate the supportive comments and thorough review of our manuscript.

The authors initially demonstrated an age-associated shift of the microbiota profile. They then moved onto showing that microbiota transfer (MT) from adult to ageing mice improved GC reactivity. Parallel experiments however, also showed that the effect of MT from aged donors into young/adult recipients surprisingly had a similar effect, thus showing that the effects on GC reactivity is independent of the age of the donor and its composition. Somehow this seems to argue against a direct role of microbiota on GC reactivity. Equally intriguing is the finding that cross-species MT does not affect GC

reactivity at all. In my opinion these are the most important observations reported here; however, the authors fell short of discussing the potential implications thoroughly.

We have revised the manuscript to ensure that these points are addressed in more detail. Many of the experiments suggested by the reviewers have helped develop the manuscript further on these points. These are outlined in the response to specific queries.

The authors also state that this finding can help to design novel microbe-based approach to sort out age-related disturbances (possibly of immune nature?). However, I do not see how. Do they envisage that transferring microbiota from young or ageing donors (they demonstrated that the effect of MT is independent of the donor's age) to ageing recipients may improve their intestinal immunity? The data do not support this conclusion; indeed, it would appear from their data that MT from individuals not genetically-matched would not work. I think that these results should be given the appropriate attention.

We appreciate the thoughtful comments from the reviewer on this point. In light of these points, and the extra knowledge gained from data generated for revision of this manuscript, we agree with the reviewer and have toned down the claims that these findings may be a suitable treatment strategy for age-related pathologies. Of note, it has previously been shown that transfer of faecal bacteria from adult into aged mice increases protection against *C. difficile* infection [manuscript reference 33], and FMT is currently used clinically in humans to treat *C. difficile* infection. Together this suggests that FMT may be a way to enhance intestinal immunity in older people.

Reference 33: Shin, J. H. *et al.* Innate Immune Response and Outcome of Clostridium difficile Infection Are Dependent on Faecal Bacterial Composition in the Aged Host. *J. Infect. Dis.* **217**, 188–197 (2018)

Also, in order to claim that MT can help to restore a young-like immune system the authors should also provide data on the effect of MT on the business end of the immune system. Several reports have shown that ageing is characterized by a sharp decline of IgA-mediated response. I suggest to evaluate mucosal (and systemic) Ab response to orally administered antigen in post-FMT mice. Alternatively, they should provide at least data on lamina propria IgA-producing plasma cells and IgA levels in intestinal washings.

To determine whether FMT influences mucosal and/or systemic antibody responses in aged mice upon oral administration of foreign antigen, we immunized 23-month-old mice with NP-conjugated cholera toxin (NP-CTx) by oral gavage, either with or without FMT. Our results show that there is no change in NP- or CTx-reactive IgA in the faecal contents, nor a change in NP- or CTx-reactive IgG1 in the serum. However, we also observed that treatment with NP-CTx alone was sufficient to boost the GC reaction in the PPs of aged mice, and that addition of FMT to NP-CTx immunization did not further boost the GC, likely accounting for the similarity of antibody titres in these mice. This result suggests that treatment of aged mice with CTx, an antigen that is highly immunogenic, can also boost the GC reaction in aged mice. This is an important observation in the context of this paper, as it demonstrates that the mucosal GC response can respond to potent antigens, which are far more amenable for therapeutic

use than FMT. These data have been included as Figure 8, and are discussed in the manuscript on page 11. The figure and text are below:

“At peripheral sites in the body, a GC reaction is normally induced in response to foreign antigen. To determine whether FMT can boost the mucosal immune response to foreign antigen, we immunised 23-month-old C57BL/6 mice with cholera toxin coupled to the hapten NP (NP-Ctx) by oral gavage three times at weekly intervals, either with or without prior FMT from 3-month-old donors. Assessment of mucosal antibody responses in the faecal contents of the ileum showed that there was no difference in anti-CTx and anti-NP IgA titres between mice that received FMT and PBS controls (Figure 8A, B). Further, there was no difference in the titre of high-affinity anti-NP2 IgA or in the ratio of NP2/NP20 binding antibodies (Figure 8 C, D), a measure of affinity maturation. Consistent with this, serum anti-CTx and anti-NP IgG1 antibody titres and affinity were not influenced by FMT in aged mice (Figure 8 E-H). Assessment of the GC reaction in the PPs of these animals, surprisingly, showed that immunisation with NP-CTx alone was sufficient to enhance the GC response in 23-month-old animals independently of FMT, with no changes in Tfh or Tfr cell number (Figure 8I-N). This indicates that FMT, like the potent immunogen CTx, might boost the GC reaction in an adjuvant-like manner to rescue the diminished GC reaction in the PPs of aged mice.”

Figure 8: Faecal microbiota transplantation (FMT) does not enhance NP-CTx-specific immune responses in the gut. 21-month-old C57BL/6 mice received FMT or PBS by oral gavage as described above, followed by three oral immunisations with NP-CTx once a week. After three weeks, NP-CTx-specific antibody levels were analysed in PBS-gavaged, NP-CTx-immunised 23-month-old C57BL/6 mice (CTx) compared to aged mice receiving FMT plus NP-CTx (FMT+CTx). Peyer's patch (PP) germinal centre (GC) responses in these mice were compared to naïve, 23-month-old C57BL/6 mice (Ctrl). **(A-D)** Antigen-specific IgA levels against CTx **(A)** and NP **(B-D)** in faecal contents from the ileum were assessed by ELISA. NP20-specific IgA **(B)**, high-affinity NP2-specific IgA **(C)** and the ratio of NP2-to-NP20-specific IgA antibodies **(D)** were used as a measure for affinity maturation. **(E-H)** Antigen-specific IgG1 levels against CTx **(E)** and NP **(F-H)** in the serum were assessed by ELISAs for CTx-specific IgG1 **(E)**, NP20-specific IgG1 **(F)**, high-affinity NP2-specific IgG1 **(G)** and the ratio of NP2-to-NP20-specific IgG1 antibodies **(H)**. **(I-N)** The percentage and number of B220⁺Ki67⁺Bcl6⁺ GC B cells **(I, J)**, CD4⁺Foxp3⁺CXCR5⁺PD1⁺ Tfh cells **(K, L)** and CD4⁺Foxp3⁺CXCR5⁺PD1⁺ Tfr cells **(M, N)** in PPs were quantified by flow cytometry. Bar plots show the combined results of 2 independent experiments with a total of n=10-13 mice per group. Bar height corresponds to

the mean, and each circle represents one biological replicate. P-values were determined using Mann-Whitney tests (**A-H**) or the Kruskal-Wallis test with Dunn's multiple testing correction (**I-N**) in GraphPad Prism6.

In addition, we have added this text to the discussion on page 13 to highlight the importance of the finding that CTx can boost the GC in older mice.

"The observation that not only microbial transfer, but also the potent mucosal antigen, cholera toxin, can boost the GC response in older animals indicates that the expansion of the GC reaction by FMT is not due to reactivation of commensal-specific memory B cells that persist in aged mice after their microbiome changes. Rather it suggests that the use of strong immunogens, or potent adjuvants, can enhance GC responses in older individuals."

Reviewer #3 (Remarks to the Author):

The authors have demonstrated that the microbiome can be transferred between young and old mice and that this influences age-related changes in gut immunity, specifically the immune cell populations of the Peyer's patches including germinal centre B cells, T follicular helper and regulatory cells. These experiments are important because it has been shown that age-related changes in the gut microbiome and intestinal architecture have profound effects on the health of the host. The inclusion of two different strains of mice, which are known to have differences in aging trajectory, mucosal immunity and IgA responses is a particular strength. I am very supportive of this manuscript as it is well written, novel and exciting and there are many important findings in here which I think will be of broad interest to readers of this journal but I have listed two suggestions below that I think would increase the level of mechanistic understanding that the microbiota has on the aging immune system.

Thank you for the thorough and supportive review of our manuscript. We particularly appreciate the comments regarding the novelty and interest of this work.

Major points

1) The findings that the microbiome alter GC B, Tfh, Tfr numbers is really exciting and I expect that this has broad implications for infections and homeostatic interactions with gut microbes. One experiment I would really like to see is whether these changes in cell numbers alter germinal centre function, perhaps in the context of an infectious disease challenge (Cdiff being of particular relevance to the elderly?) or even with regard to the IgA repertoire.

It has previously been reported that transfer of faecal bacteria from young mice into aged mice increases protection against *C. difficile* infection. This is included in the last paragraph of the discussion of this manuscript to highlight the importance of this treatment to improving health outcomes in older individuals.

Reference 33: Shin, J. H. *et al.* Innate Immune Response and Outcome of Clostridium difficile Infection Are Dependent on Faecal Bacterial Composition in the Aged Host. *J. Infect. Dis.* **217**, 188–197 (2018)

The last paragraph of the discussion reads:

"The transfer of a young microbiome into aged mice increases protection against *C. difficile* infection ³³, indicating that the microbiota of young animals can functionally boost intestinal immune protection. This makes the gut microbiome a possible target for the treatment of a range of age-associated symptoms. FMT ⁴¹, probiotics ⁴⁰, co-habitation ⁴² and diet ⁴³ all have an impact on the composition of the gut microbiome and could prove to be innovative interventions to facilitate healthy ageing."

As the authors state in the introduction, since IgA shapes the microbiome, could age-related changes in the GC be cause or consequence of dysbiosis? For example, for species lost during the course of aging (and restored by microbiome transfer), is there a loss of IgA that is restored during the microbiome transfer (and could this be detected by a bacterial flow cytometry experiment?). Presumably measures of IgA could be taken in a non-invasive manner so that they could be measured before and after transfer.

To determine whether ageing or microbial transfer influences IgA antibody responses in the gut, we first compared IgA coating of gut bacteria between young and aged mice. We did not observe an age-dependent change in the frequency of IgA-coated bacteria in either C57BL/6 or BALB/c mice. These data have been included in the manuscript as Supplementary Figure 4 and described in the text on page 6. Consistent with this, there was no restoration in IgA coating after cohousing or after FMT, and these data have been included in the manuscript as Supplementary Figures 5 and 7 and described in the text on page 6 and 9, respectively. The revised text and figures are below:

Page 6 in the manuscript:

"To determine whether there is a link between the poor GC reaction in the PPs of aged mice and antibody regulation of the microbiome, we examined IgA coating of gut bacteria in aged mice. In both the ileum and colon the frequency of IgA coated bacteria was comparable between 3-month-old adult and 22-month-old aged mice from both the C57BL/6 and BALB/c strains (Supplementary Figure 4). This could be explained by previous studies which described normal levels of intestinal IgA and commensal IgA coating in GC-deficient mice, suggesting that commensal-specific IgA can be produced in a GC-independent manner ²⁷⁻²⁹. Taken together, this indicates that the impaired GC reaction in the PPs of aged mice does not influence commensal reactive IgA, and prompts the hypothesis that the composition of the microbiome may influence the magnitude of the GC response."

Supplementary Figure 4: IgA-coating of faecal bacteria is not affected by ageing. Bacterial IgA-coating were assessed in adult (3-month-old; 3mo) and aged (22-month-old; 22mo) mice. **(A)** Gating strategy for IgA-coated bacteria. **(B, C)** Representative flow cytometric plots of IgA-coated faecal bacteria isolated from the ileum **(B)** and colon **(C)** of BALB/c mice. **(D-G)** Quantitation of IgA-coating of bacteria in faecal contents isolated from the ileum **(D, E)** and colon **(F, G)** of adult and aged BALB/c **(D, F)** and C57BL/6 **(E, G)** mice. Bar plots show the combined results of 2-4 independent experiments with a total of n=11-16 mice per group. Bar height corresponds to the mean, and each circle represents one biological replicate. P-values were determined using the Mann-Whitney test in GraphPad Prism6.

“To our surprise, the age-associated reduction of GC B cells in PPs of BALB/c mice was lost upon such co-housing (Figure 3A, B). This correction of the GC reaction was accompanied by an increase in Tfh cells but Tfr cell numbers and IgA coating of commensals were unchanged (Figure 3C-F; Supplementary Figure 5).”

Supplementary Figure 5: Co-housing does not affect IgA-coating of faecal bacteria. Bacterial IgA-coating were assessed in adult (3-4-month-old; 3mo/4mo) and aged (22-month-old; 22mo) mice after co-housing for 40 days. **(A-D)** Quantitation of IgA-coating of bacteria in faecal contents isolated from the ileum **(A, B)** and colon **(C, D)** of cohoused BALB/c **(A, C)** and C57BL/6 **(B, D)** mice. Bar plots show the combined results of 2-4 independent experiments with a total of n=19-30 mice per group. Bar height corresponds to the mean, and each circle represents one biological replicate. P-values were determined using the Mann-Whitney test in GraphPad Prism6.

Page 9 in the manuscript:

“To determine whether FMT can rescue the PP GC reaction in aged mice of another strain, we performed FMT experiments in 22-month-old BALB/c mice (experimental set-up as in Figure 5A). GC B and Tfh cells had significantly expanded in 22-month-old mice 23 days after FMT compared to PBS-treated control mice (Figure 6A-D), but Tfr cells were not changed by FMT treatment (Figure 6E, F). FMT did not affect the levels of PP-resident IgA⁺ B cells, free intestinal IgA or bacterial IgA coating in either mouse strain (Supplementary Figure 7).”

Supplementary Figure 7: Faecal microbiota transplantation (FMT) does not affect gut IgA responses. 21-month-old mice were given a suspension of faecal pellets taken from 3-month-old mice by oral gavage. The cages of these aged recipients were supplemented with fresh faecal pellets and dirty bedding from these donors once a week. A control group received PBS by oral gavage. After 3 weeks, intestinal IgA levels were assessed in adult donor mice (4mo), aged control mice (23mo+PBS) and aged mice receiving FMT (23mo+FMT). **(A)** Representative flow cytometric plots for IgA⁺ B cells (B220⁺IgA⁺) cells in the Peyer's patches (PPs) of adult (3-month-old; 3mo) and aged (22-month-old; 22mo) BALB/c mice. **(B-E)** Quantitation of IgA⁺ B cells in percentage **(B, D)** and cell numbers **(C, E)** in BALB/c **(B, C)** and C57BL/6 **(D, E)** mice by flow cytometry. **(F-I)** Quantitation of faecal IgA in faecal contents isolated from the ileum **(F, H)** and colon **(G, I)** of BALB/c **(F, G)** and C57BL/6 **(H, I)** mice by ELISA. **(J-M)** Quantitation of IgA-coating of bacteria in faecal contents isolated from the ileum **(J, L)** and colon **(K, M)** of BALB/c **(J, K)** and C57BL/6 **(L, M)** mice. Bar plots show the combined results of 2 independent experiments with a total of n=8-16 mice per group. Bar height corresponds to the mean, and each dot represents one biological replicate. P-values were determined using the Kruskal-Wallis test with Dunn's multiple testing correction in GraphPad Prism6.

2) There is a lot of data in the heterochronic co-housing to think through. For example, the aged BALB/c mice appear to have GC B cells restored (no effect on Tfr) but enhanced Tfh whereas the aged C57bl/6 mice may or may not changes in GC B cells (depending on whether you look at % or #), no effect on age related increases in Tfh and maintain the higher Tfr with age but clearly do have altered microbiota. The experiments in Fig 5 add complexity to this as they imply that any microbiome transfer can increase proportion of all three cell types irrespective of the age of the host. I wonder if the authors have considered whether the (apparent) discrepancy btw other experiments is due to the difference in measuring at 3 wks or 40 days? (ie. Could some of these increased cell numbers be transient). The control here would be the young-young mice or old-old mice to disentangle age of the host versus the process of co-housing (especially since the data in Fig 7 demonstrate that in young

mice with an intact microbiota, there are no changes, increasing the evidence that this is microbiome independent).

It is a good point that the boost of the GC response in aged mice may be transient, and account for differences in the GC magnitude in the FMT and co-housing experiments. This point has been included in the discussion on page 13. The text is below:

“Interestingly, the GC reaction in PPs was more strongly boosted following FMT than by co-housing. If the increase in magnitude of the GC response in aged mice is transient, this discrepancy could be explained by the duration of the different experiments, as we assessed the PPs GC reaction 23-days after FMT, and 40 days after the start of cohousing.”

In addition, we performed an experiment in which we co-housed 3-month-old adult animals from different strains (C57BL/6 and Balb/c) and did not observe a change in percentage or number of GC B cells, Tfh cells or Tfr cells. These data are presented below:

Figure for reviewer: Cross-strain co-housing does not affect the germinal centre (GC) response in Peyer’s patches (PPs). Adult BALB/c and C57BL/6 mice were co-housed for 40 days, then Peyer’s patch (PP) germinal centre (GC) cell populations were analysed by flow cytometry. The percentage and number of $B220^+Ki67^+Bcl6^+$ GC B cells (A, B), $CD4^+Foxp3^-CXCR5^+PD-1^+$ Tfh cells (C, D) and $CD4^+Foxp3^+CXCR5^+PD-1^+$ Tfr cells (E, F) in Peyer’s patches are shown. Bar plots show the combined results of one experiment with n=7 mice per group. Bar height corresponds to the mean, and each circle represents one biological replicate. No significant differences were found between groups by Kruskal-Wallis tests with Dunn’s multiple testing correction in GraphPad Prism6.

3) In understanding the functional consequences of age related changes in the GC, I'm reminded of this paper <https://doi.org/10.1016/j.immuni.2015.08.011> describing strain specific differences in IgA, independent of the microbiome. Is it possible that the BALB/c strain is more susceptible to age-related changes in GC? Would this be apparent in a challenge model or in IgA load in the serum or faecal pellets?

In general, we do observe that the age-associated changes in the immune system are more pronounced in the BALB/c than the C57BL/6 strain, and BALB/c mice have higher levels of free, faecal IgA than C57BL/6 (as previously reported). However, further experiments prompted by revision of this paper did not show a difference in IgA coating of gut bacteria by ageing or between strains (Supplementary Figures 4 and 5, above in response to this reviewer's first point), suggesting that strain-specific differences in IgA may not play a role in the age-related changes in the GC reaction.

Minor points

1) typo in the introduction 'Those GC B cell which able to ...' = "Those GC B cell which*are* able to ..."

This typo has been corrected.

2) I know that taxa plots are a bit pedestrian but In Fig 2G & H I found that there were a few gems that are hard to discern in the Krona plots. For example, the Bacteroidales/Firmicutes ratios have been shown to change with age before in both mice and humans (and it is very valuable to the community to demonstrate this is a reproducible change!) and I think that this is much more exaggerated in the BALB/c is interesting (could the Firmicutes in the C57 be coloured green as well for ease of comparison?). I'm also confused about 'uncultured' as a classification – were these 'unclassified' perhaps (I doubt the authors cultured all the strains). Beside the 3% Roseburia in the 3mo BALBC there seems to be an unlabelled group? Some of these gems might be more obvious is taxa plots representing order, genus etc.

We have represented the data as taxa plots which are included in Supplementary Figure 3. We have also changed 'uncultured' to 'unclassified' as this is the correct notation. The figure is below, and is called out in the manuscript along with the Krona plots.

Supplementary Figure 3: Age-associated changes in the taxa composition of the gut microbiome in C57BL/6 and BALB/c mice. 16S rRNA sequencing data were generated from

faecal pellets collected from five adult (3-month-old; 3mo) and five aged (21-month-old; 21mo) female BALB/c mice as well as male and female C57BL/6 mice. **(A-D)** Age-associated changes in the gut microbiome of female C57BL/6 mice. **(A)** Bray-Curtis PCoA of samples collected from female C57BL/6 mice. **(B)** Shannon diversities of samples collected from female C57BL/6 mice. The p -value was generated from an ANOVA test. **(C)** Depiction of bacterial families whose abundance was significantly different between adult and aged female C57BL/6 mice as determined by ANOVA analysis after cumulative-sum scaling (CSS). *FDR ≤ 0.05 , **FDR ≤ 0.01 , ***FDR ≤ 0.001 . **(D)** Krona plots depicting the phylogenetic composition of the gut microbiome in 3-month-old (left) and 21-month-old (right) C57BL/6 females. The percentages shown are averages of the samples in each age group. **(E-G)** Taxa plots of bacterial phyle **(E)**, orders **(F)** and families **(G)** detected in faecal samples from adult (3month-old) and aged (21-22month-old) BALB/c and C57BL/6 mice generated in QIIME2 with a total of $n=5-10$ mice per group. In **(F)** and **(G)** only the 15 most abundant bacterial orders/families are listed in the legend.

Best of luck with an interesting manuscript.

Thank you!

Reviewers' comments:

Reviewer #1 (Remarks to the Author):

The authors have adequately addressed my concerns raised in my first review of this submission and I now feel that it is ready for publication.

Reviewer #2 (Remarks to the Author):

The authors have presented a significantly revised version of the ms and provided further evidence that the gut microbiome affected the GC reaction. The authors should be commended for it.

Of note is the notion that this effect was not accompanied by modifications of the IgA responses. Thus, it would seem that the observations reported here represent an interesting notion with no potential application in order to "rejuvenate" the aging adaptive immune responses as claimed by the authors. Also, it is not clear as to why microbiota from a genetically different mouse strain that harbor a different microbiota failed to improve GC reactivity. The finding that cross-species FMT does not have any effect on CG reaction. The "young" microbiota differed from the "aged" one; this scenario resembled closely the difference between the two mouse strains used here; however the microbiota from young syngenic mice did modify GC reactivity in aged recipients while the cross-species FMT did not. This finding is also intriguing given the notion that a different stimulus of bacterial origin (cholera toxin) does so. At least, this should have compelled the authors to change the title of the article, including a clear reference to the cholera toxin results and drop the hint of the microbiota-specific role.

Overall, the authors fell short of discussing this important aspect of the story; instead they keep interpreting their results as a clear-cut evidence to support the effects of the gut microbiome on GC reactivity. Yet, how comes that the aged microbiota triggered an improved GC reactivity in young recipients? If what is described here is due specifically to the microbiota, why does the aged microbiota fail to maintain CG reactivity in the aged mouse while it restored it in young recipients? Also, reference 36 is not completely appropriate to support the statement that gut microbiome has as an impact on immunity. In that article the authors focused on innate immunity, a rather different type of immune response.

Reviewer #3 (Remarks to the Author):

The authors have done a substantive amount of work to address all three reviewer's (refreshingly consistent) comments and questions. At the end of the day it is not completely apparent what the biological significance of the age-microbiome dependent changes in the germinal centres mean; however, I am supportive of publication of this work for a number of reasons. 1) It is an exhaustive examination of gut immunity in the context of age & the microbiome and fills a number of important knowledge gaps in mucosal immunology of the aging gut and 2) it begins the process of unravelling the chicken/egg question of what comes first microbial dysbiosis or loss of immune function (I would have guessed the reduction in IgA due to age-related loss of GC would have led to dysbiosis and I would have been wrong). I am supportive of this manuscript.

Reviewers' comments:

Reviewer #1 (Remarks to the Author):

The authors have adequately addressed my concerns raised in my first review of this submission and I now feel that it is ready for publication.

We thank the reviewer for their review.

Reviewer #2 (Remarks to the Author):

The authors have presented a significantly revised version of the ms and provided further evidence that the gut microbiome affected the GC reaction. The authors should be commended for it.

Thank you for the supportive comments

Of note is the notion that this effect was not accompanied by modifications of the IgA responses. Thus, it would seem that the observations reported here represent an interesting notion with no potential application in order to "rejuvenate" the aging adaptive immune responses as claimed by the authors.

We have removed the word rejuvenate from the title of the manuscript, and removed all references to this in the paper. Together with addressing the comments below the new title now reads "Faecal transplantation or a bacterial toxin boosts gut germinal centres in aged mice"

Also, it is not clear as to why microbiota from a genetically different mouse strain that harbor a different microbiota failed to improve GC reactivity the finding that cross-species FMT does not have any effect on CG reaction. The "young" microbiota differed from the "aged" one; this scenario resembled closely the difference between the two mouse strains used here; however the microbiota from young syngenic mice did modify GC reactivity in aged recipients while the cross-species FMT did not. This finding is also intriguing given the notion that a different stimulus of bacterial origin (cholera toxin) does so. At least, this should have compelled the authors to change the title of the article, including a clear reference to the cholera toxin results and drop the hint of the microbiota-specific role. Overall, the authors fell short of discussing this important aspect of the story; instead they keep interpreting their results as a clear-cut evidence to support the effects of the gut microbiome on GC reactivity. Yet, how comes that the aged microbiota triggered an improved GC reactivity in young recipients? If what is described here is due specifically to the microbiota, why does the aged microbiota fail to maintain CG reactivity in the aged mouse while it restored it in young recipients?

We agree with the reviewer that it is surprising that cross-strain FMT does not boost the GC while FMT between mice of different ages from the same strain does not. While this was not the result we expected, we do not know the mechanism underpinning this, nor is there any literature that explains the inability of cross-strain FMT to enhance the GC response.

However, there are some possibilities that may contribute to this, and we have included an extra text in the results and an additional paragraph in the discussion to consider these. The text is below.

Results:

“This suggests that the GC reaction in PPs responds specifically to heterochronic faecal transplantation and that host genetics might impact the cross-talk between the gut microbiota and the PP GC reaction.”

Discussion:

“Cross-strain FMT from BALB/c into C57BL/6 did not boost the GC reaction. This is surprising, as the differences in gut microbial composition between C57BL/6 and BALB/c mice are similar to the differences observed between adult and aged mice in which FMT does enhance PP GC responses. While we have not been able to dissect the mechanism behind this, host genetics might play a role in shaping the cross-talk between the gut microbiome and PP GC responses. Consistent with existing literature, we observed clear differences in mucosal IgA levels between BALB/c and C57BL/6 mice (Suppl. Figure 7 F-I)³³. Fransen *et al.* linked reduced levels of mucosal IgA in C57BL/6 mice with their impaired ability to mount antigen-specific IgA antibody responses to a non-invasive bacterial species³³. Thus, the genetic predisposition of C57BL/6 mice for reduced IgA production might impair their ability to mount PP GC responses to BALB/c-derived commensals.”

To make it clear that increasing the magnitude of the GC response in aged mice is not limited to the microbiota we have changed the abstract, introduction, discussion and the title to highlight the results from the cholera toxin immunisation experiments. We hope that it is now clear in the manuscript that another stimulus, of bacterial origin, is able to increase the magnitude of the GC response in aged mice. The altered sections of the manuscript are included below.

Title:

“Faecal transplantation or a bacterial toxin boosts gut germinal centres in aged mice”

Abstract:

“Here, we establish that the defective germinal centre reaction in Peyer’s patches of aged mice can be rescued by co-housing of adult and aged mice, and via faecal transfers from younger adults into aged mice and by immunisations with cholera toxin. This demonstrates that the poor germinal centre reaction in aged animals is not irreversible, and that it is possible to improve this response in older individuals by providing appropriate stimuli.”

Introduction:

“Here we report that the defective GC reaction in aged mice could be boosted by co-housing with younger animals, by direct faecal transplantation from adult donors and by oral administration of cholera toxin. This demonstrates that the age-dependent defect in the gut GC reaction is not irreversible, but can be corrected by changing the microbiota or by delivery of a bacterial derived toxin.”

Discussion:

“These data demonstrate that the defective germinal centre response in aged mice is not a cell intrinsic feature of the ageing immune system and can be restored by replenishment of the microbiome or stimulation with cholera toxin.”

“Our data suggest that, in the PPs of aged mice, the capacity of GC B cells to respond to antigen is not impaired in a cell-intrinsic manner, but can be rescued by stimulation from the microbiota or cholera toxin.”

Also, reference 36 is not completely appropriate to support the statement that gut microbiome has as an impact on immunity. In that article the authors focused on innate immunity, a rather different type of immune response.

We have changed the wording of this sentence in the discussion to make it clear that this paper referred to boosting the innate immune system.

“Further, the transfer of a young microbiome into aged mice increases protection against *C. difficile* infection by stimulating the innate immune system³⁷, indicating that the microbiota of young animals can functionally boost intestinal immune protection.”

Reviewer #3 (Remarks to the Author):

The authors have done a substantive amount of work to address all three reviewer's (refreshingly consistent) comments and questions. At the end of the day it is not completely apparent what the biological significance of the age-microbiome dependent changes in the germinal centres mean; however, I am supportive of publication of this work for a number of reasons. 1) It is an exhaustive examination of gut immunity in the context of age & the microbiome and fills a number of important knowledge gaps in mucosal immunology of the aging gut and 2) it begins the process of unravelling the chicken/egg question of what comes first microbial dysbiosis or loss of immune function (I would have guessed the reduction in IgA due to age-related loss of GC would have led to dysbiosis and I would have been wrong). I am supportive of this manuscript

We thank the reviewer for their supportive comments.